# Tristetraprolin attenuates schistosomiasis-induced liver fibrosis through m⁶A-mediated regulation of TGF-β1 mRNA stability

Xuejun Zhao[1,2,3], Xinyi Wang[1,2], Kejun Liu[1,2], Qian Fang[1,2], Cheng Wang[1,2], Guangbo Mei[1,2], Lihua Qu[4], Jiaxi Zhang[1,2], Jiayi Feng[1,2], Wenjuan Yang[1,2], Junnan Zhu[1,2], Hang Zhuo[1,2], Na Kuang[5], Xuebing Qiu[6], Zhigao Deng[7], Qiong Luo[8], Xiaoxia Jin[9], Lanxuan Jiang[10], Xiaoqing Li[11]*, Huifen Dong[1,2]*, Rui Zhou 🄳[1,2]*

**1** Hubei Province Key Laboratory of Allergy and Immunology, School of Basic Medical Sciences, Wuhan University, Wuhan, Hubei, People's Republic of China, **2** Department of Medical Parasitology, School of Basic Medical Sciences, Wuhan University, Wuhan, Hubei, People's Republic of China, **3** Department of Clinical Laboratory, Guizhou Provincial People's Hospital, Guiyang, Guizhou, People's Republic of China, **4** School of Basic Medical Sciences, Hubei University of Science and Technology, Xianning, Hubei, People's Republic of China, **5** Ecological Station of Oncomelania Hupensis in Gong'an County, Jingzhou, Hubei, People's Republic of China, **6** Wuhan Kindstar Diagnostics Co Ltd, Wuhan, Hubei, People's Republic of China, **7** Zhongnan Hospital of Wuhan University, Institute of Hepatobiliary Diseases of Wuhan University, Transplant Center of Wuhan University, National Quality Control Center for Donated Organ Procurement, Hubei Key Laboratory of Medical Technology on Transplantation, Hubei Provincial Clinical Research Center for Natural Polymer Biological Liver, Wuhan, Hubei, People's Republic of China, **8** Department of Anesthesiology, Zhongnan Hospital of Wuhan University, Wuhan, Hubei, People's Republic of China, **9** Department of Parasitology, Bengbu Medical University, Bengbu, Anhui, People's Republic of China, **10** Department of Pharmacology, School of Basic Medical Sciences, Wuhan University, Wuhan, Hubei, People's Republic of China, **11** Center for Stem Cell Research and Application, Union Hospital, Tongji Medical School, Huazhong University of Science and Technology, Wuhan, Hubei, People's Republic of China

* lxqtom@aliyun.com (XL); hfdong@whu.edu.cn (HD); ruizhou@whu.edu.cn (RZ)

## Abstract

Inhibiting the activation of hepatic stellate cells (HSCs) represents a key therapeutic strategy for alleviating liver fibrosis induced by schistosomiasis. Diverse cell populations secrete pro-inflammatory cytokines and chemokines, which induce HSC activation and thereby promote hepatic fibrosis progression. Tristetraprolin (TTP) exerts a pivotal role in the post-transcriptional regulation of pro-inflammatory cytokines by either accelerating mRNA degradation or suppressing translation, a regulatory mechanism closely associated with the pathogenesis of various hepatic disorders. However, the pathological roles of TTP in *Schistosoma japonicum*-induced liver fibrosis remain largely elusive. Here, we report that TTP is upregulated in the liver during *S. japonicum*-induced liver fibrosis, and its overexpression markedly ameliorates this fibrotic pathology *in vivo*. We further identify that TTP negatively regulates TGF-β1 mRNA stability by promoting N6-methyladenosine (m⁶A) RNA methylation, thereby inhibiting HSC activation. Mechanistically, TTP enhances transcription of the WT1-associated protein (*WTAP*) gene through its interaction with SMAD2/3. Furthermore, treatment with an m⁶A RNA methylation inhibitor confirms that TTP-mediated

**Data availability statement:** MeRIP Seq data generated in this study have been deposited to the GEO database under accession number GSE272777. Published RNA seq datasets were obtained from GEO under accession number GSE157581. These datasets were used for the analyses presented in Figs 4B and S5A. Published RNA-seq datasets were obtained from the GEO database under accession numbers GSE59276 and GSE61376, and were used for the analyses presented in Figs 1B and S1A–S1C. The proteomics data generated in this study have been deposited to the ProteomeXchange Consortium under the identifier PXD056357, and were used for the analysis presented in Fig 6B.

**Funding:** This work was supported by the National Natural Science Foundation of China (NSFC) [No. 82172301 to HD, No. 82271795, 81771702 to RZ], Science and Technology Foundation Project of Guizhou Health Commission [gzwkj2025-616 to XJ.Z], Talent Fund of Guizhou Provincial People's Hospital [Talent Project of the Institute [2023]-43 to XJ.Z], and Natural Science Foundation of Hubei Province, China (2024AFB726 to QL). The funders had no role in study design, data collection and analysis, decision to publish, or preparation of the manuscript.

**Competing interests:** The authors have declared that no competing interests exist.

protection against *S. japonicum*-induced liver fibrosis, an effect associated with increased m⁶A RNA methylation *in vivo*. Thus, our findings uncover a novel and critical role of TTP in exerting its anti-fibrotic function via the WTAP/m⁶A epitranscriptomic machinery in the pathogenesis of *S. japonicum*-induced liver fibrosis. This finding provides a rationale for the therapeutic targeting of TTP-mediated m⁶A RNA methylation in *S. japonicum*-induced liver fibrosis.

## Author summary

Activation of hepatic stellate cells (HSCs) represents a pivotal event in the progression of schistosomiasis-induced liver fibrosis. Pro-inflammatory cytokines and chemokines secreted by injured hepatocytes and activated macrophage induce HSC activation. However, the underlying mechanisms governing the fine-tuned regulation of pro-inflammatory cytokines and chemokines gene expression during the progression of schistosomiasis-induced liver fibrosis remain largely elusive. As a key negative regulator of immune responses, the role of TTP in liver fibrosis remains controversial. Furthermore, while TTP has been reported to act as a transcriptional co-regulator and modulate gene expression at the transcriptional level, the majority of TTP-related studies have centered on its canonical role as a key mediator of mRNA degradation. Here, our study demonstrates that TTP attenuates *S. japonicum*-induced liver fibrosis via the regulation of TGF-β1 mRNA stability in an m⁶A-dependent manner. Specifically, WTAP-mediated m⁶A RNA methylation of TGF-β1 mRNA exerts a suppressive effect on HSC activation, thereby mitigating *S. japonicum*-induced liver fibrosis. Notably, our findings uncover a previously uncharacterized regulatory mechanism whereby TTP promotes *WTAP* transcription via direct interaction with SMAD2/3. These results demonstrate that TTP-driven transcriptional regulation of WTAP represents a novel mechanism underlying HSC activation, and further identify a potential strategy for intervening in *S. japonicum*-induced liver fibrosis.

## Introduction

Schistosomiasis is a neglected endemic and neglected tropical disease caused by *Schistosoma* parasites, affecting more than 200 million people across 78 countries [1]. *Schistosome* eggs release bioactive molecules that provoke robust host immune responses, leading to hepatic fibrosis that may ultimately progress to cirrhosis. Hepatic fibrogenesis is driven by paracrine signaling from injured hepatocytes and activated macrophages, leading to the activation of resident hepatic stellate cells (HSCs). Upon activation, these HSCs differentiate into myofibroblasts that produce extracellular matrix proteins at a rate exceeding their degradation [2,3]. Accordingly, portal hypertension secondary to hepatic fibrosis constitutes a leading cause of morbidity and mortality in patients with schistosomiasis [4].

The activation of hepatic stellate cells (HSCs) is a pivotal event driving schistosomiasis-induced liver fibrosis progression. Several pro-fibrotic factors secreted by damaged hepatocytes and activated macrophages—notably TGF-β1—directly induce HSC activation, with TGF-β1 acting as the master regulator of this fibrosis-underpinning core process [5,6]. MYB is a well-characterized transcriptional activator of pro-fibrotic genes in HSCs, driving hepatic fibrogenesis [7]. Tristetraprolin (TTP, also known as ZFP36), a critical RNA-binding protein, contains several tandem CCCH zinc finger domains. It binds to AU-rich elements (AREs) within 3′-untranslated regions (3′UTRs) of target mRNAs to promote their degradation and/or prevents their efficient translation [8]. It has also been reported that TTP negatively regulates several inflammatory cytokines/chemokines, such as interleukin 6 (IL-6), interleukin 8 (IL-8), and C-C motif chemokine ligand 2 (CCL2) [9]. Accordingly, these regulatory effects of TTP are closely associated with the pathogenic progression of multiple hepatic disorders, including acute liver failure, hepatic fibrosis, and hepatocellular carcinoma (HCC) [10,11]. However, the pathological roles of TTP in hepatic fibrosis remain controversial. TTP has been reported to promote apoptosis and inhibit HSC activation and invasion *in vitro* [10]. Conversely, another study demonstrated that TTP exacerbates hepatic steatosis, inflammation, and fibrosis in mice maintained on a methionine/choline-deficient diet [11]. Collectively, the precise role of TTP in the pathogenesis of hepatic fibrosis remains to be further elucidated.

N6-methyadenosine (m6A) methylation of RNA represents the most prevalent epitranscriptomic modification of RNA. This reversible epigenetic process is dynamically regulated by the coordinated action of m6A "writers" (core methyltransferases METTL3, METTL14, and WTAP), "erasers" (demethylases ALKBH5 and FTO), and "readers" (YTH domain-containing effector proteins YTHDF1, YTHDF2, and YTHDF3), which plays vital role in determining the fate of target RNAs [12,13]. Additionally, certain zinc finger proteins have also been reported to act as key regulators of m6A RNA modification. Our previous findings have shown that TTP enhances m6A RNA modification by promoting the transcription of *WTAP*, *METTL14*, and *YTHDF2* [14]. Notably, TTP is known as a transcriptional co-regulator and can interact with various transcription factors to regulate gene expression at the transcriptional level [15]. However, the molecular mechanisms governing TTP-mediated transcriptional regulation of m6A-associated genes remains to be fully elucidated.

In this study, we elucidated the functional role of TTP and the specific mechanism by which TTP modulates the pathology progression of *S. japonicum*-induced liver fibrosis. TTP was identified to negatively regulate TGF-β1 by promoting *WTAP* transcription, thereby inhibiting HSC activation. Specifically, we demonstrated that TTP promotes *WTAP* transcription by interacting with SMAD2/3. Collectively, our study demonstrates that TTP-driven m6A RNA hypermodification is a critical regulator of schistosomiasis-related liver fibrosis pathogenesis, and provides key insights into developing targeted therapeutics for this fibrotic disorder.

## Results

### Expression level of TTP in *S. japonicum*-induced liver fibrosis

To investigate the potential functional role of TTP in liver fibrosis induced by *S. japonicum* infection, we observed an upregulation of TTP expression in a murine model of *S. japonicum*-induced liver fibrosis (Fig 1A). Interestingly, TTP expression began to increase in the liver tissues of *S. japonicum*-infected mice at 4 weeks post-infection, peaked at 8 weeks, and positively correlated with the severity of hepatic fibrosis (Fig 1A). Publicly available murine transcriptome datasets (GSE59276) validated the upregulation of *Ttp* mRNA in *S. japonicum*-infected mice (Fig 1B). In addition, published genome-wide expression profiling data from liver tissues of patients with schistosomiasis-induced hepatic fibrosis and healthy controls (GSE61376) confirmed the upregulation of TTP mRNA in fibrotic liver tissues compared to healthy controls (Fig 1B).

Using Spearman's rank correlation analysis and data from the GSE61376 dataset, we identified a significant positive correlation between *TTP* expression and the expression levels of *COL1A1*, *COL4A1*, and *TGFB1* in human liver tissues with *S. japonicum*-induced liver fibrosis (S1A–S1C Fig). Furthermore, double immunofluorescence staining of liver sections revealed that TTP primarily colocalized with albumin (a hepatocyte-specific marker) in *S. japonicum*-infected liver tissues

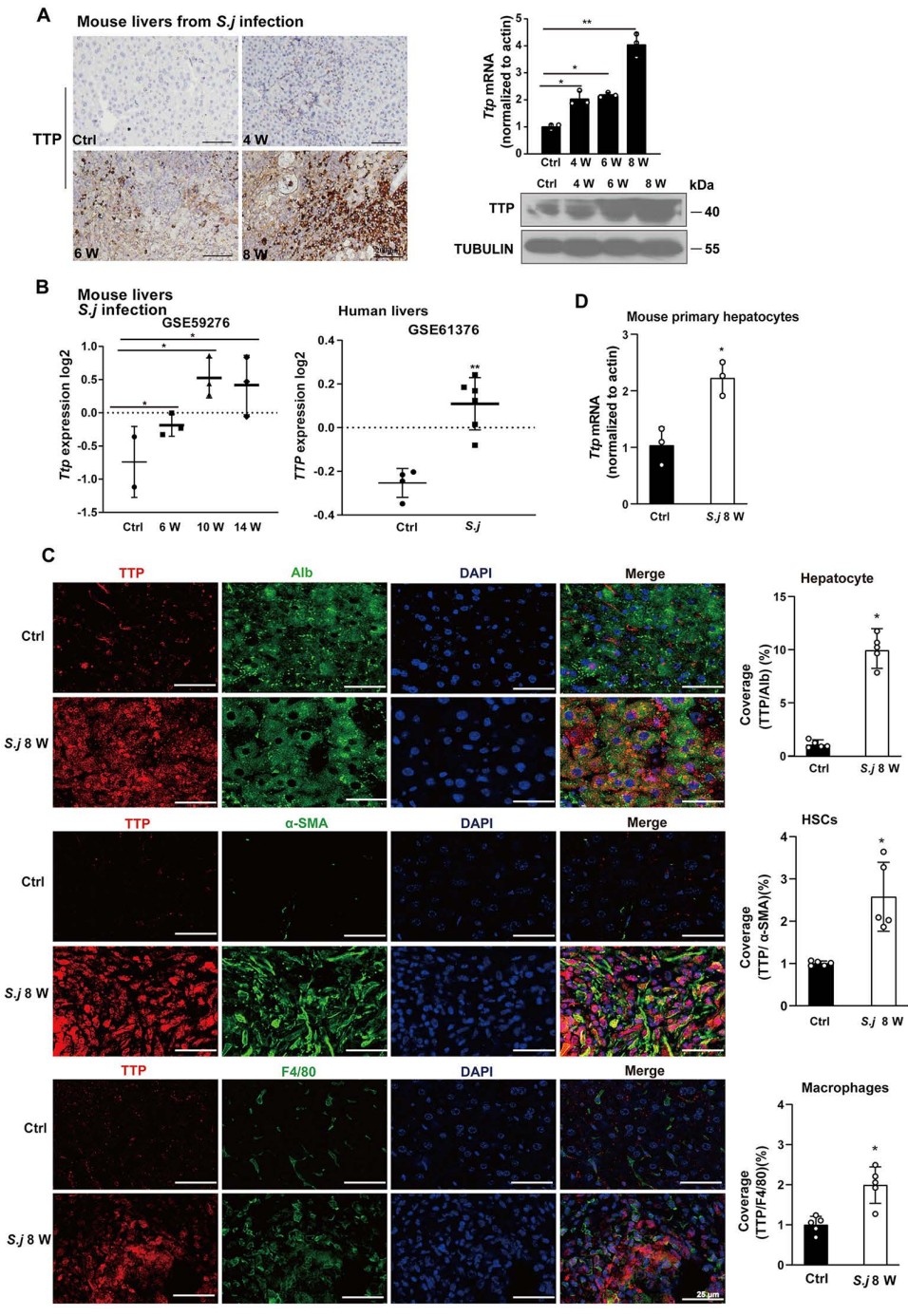

**Fig 1. TTP expression is increased during *S. japonicum*-induced liver fibrosis. (A)** Immunohistochemical (IHC) analysis, western blot, and real-time PCR were performed to evaluate TTP expression in fibrotic liver tissues from mice infected with *S. japonicum* (Scale bar: 200 μm) **(B)** Analysis of *TTP* transcript levels in mouse and human liver samples with *S. japonicum*-induced liver fibrosis, from two independent microarray datasets: GSE59276 and GSE61376. **(C)** Liver fibrosis samples of *S. japonicum*-infected mice were processed for immunofluorescence co-staining for TTP with Alb (hepatocyte marker), a-SMA (activated HSC marker), and F4/80 (macrophage marker). Representative immunofluorescence images and corresponding quantification showing TTP expression in hepatocytes, HSCs, and macrophages of liver tissues (Scale bar: 25 μm). **(D)** The expression of TTP in hepatocytes within mouse livers following *S. japonicum* infections. Data are presented as mean±SD of 3 mice per group and are representative of three independent experiments. Statistical analyses were performed using one-way ANOVA or 2-tailed Student's t-test unpaired. *$P<0.05$ versus Ctrl. **$P<0.01$ versus Ctrl.

at 8 weeks post-infection, indicating its localization in hepatocytes during fibrosis (Fig 1C). Additionally, TTP was detected in α-SMA-positive hepatic stellate cells (HSCs), a well-recognized specific marker for activated HSCs (Fig 1C). Although TTP expression was also upregulated in macrophages following *S. japonicum* infection, the magnitude of this increase was substantially lower than that observed in hepatocytes and HSCs (Fig 1C). To further characterize TTP expression pattern in distinct liver cell populations, we isolated hepatocytes (HCs), HSCs, and hepatic macrophages from the livers of naive mice. Real-time PCR analysis revealed that *Ttp* mRNA levels were significantly higher in primary HCs and HSCs compared to macrophages (S1D Fig). We also isolated primary hepatocytes from the livers of *S. japonicum*-induced fibrotic mice to analyze the TTP expression patterns. Consistent with our immunofluorescence data, *Ttp* mRNA levels were significant upregulated in primary hepatocytes from *S. japonicum*-infected mice compared to controls (Fig 1D). Additionally, a marked increase in Ttp mRNA expression was detected in primary hepatocytes following stimulation with soluble egg antigen (SEA) for 24 h (S1E Fig). Collectively, these data from patients with schistosomiasis-induced liver fibrosis and corresponding murine models demonstrate marked upregulation of TTP during the progression of liver fibrosis.

## TTP knockdown accelerates schistosomiasis-induced liver fibrosis *in vivo*

Given the elevated TTP expression in fibrotic hepatic tissues, we generated adeno-associated virus (AAV) vectors expressing a short hairpin RNA (shRNA) targeting TTP (AAV-shTTP) or a non-targeting control (AAV-shCtrl). Efficient TTP knockdown was confirmed in hepatic tissues from AAV-injected, non-infected control mice via Real-time PCR and Western blot analysis (S2A Fig). We next investigated the impact of TTP knockdown on hepatic fibrosis in *S. japonicum*-infected mice. As described in a previous study [16], mice were first infected with a mild dose of *S. japonicum* cercariae, and then injected with PBS, AAV-shTTP or AAV-shCtrl at day 10 post-infection (Fig 2A). Hematoxylin and eosin (H&E) and Masson's trichrome staining results demonstrated that AAVDJ-Ctrl injection did not affect the progression of schistosomiasis-induced liver fibrosis compared with the PBS control group, ruling out potential off-target effects of the AAV vector itself on schistosomiasis-induced liver fibrosis (S2B–S2E Fig). The expression of TTP was significantly lower in the AAV-shTTP-treated group than in the control group following *S. japonicum* infection, confirming that AAV-shTTP effectively knocked down TTP expression *in vivo* (Fig 2B). Notably, H&E staining showed that the area of egg granuloma was significantly expanded in the AAV-shTTP-treated mice after *S. japonicum* infection (Fig 2C–2D). Hepatic egg loads showed no significant difference between AAV-shCtrl and AAV-shTTP mice (Fig 2E), indicating that TTP knockdown did not affect hepatic egg deposition following *S. japonicum* infection. Furthermore, TTP knockdown had no impact on the liver weight proportion in infected mice (Fig 2E). Moreover, Masson's trichrome staining assays revealed that TTP knockdown exacerbated the severity of *S. japonicum*-induced liver fibrosis compared to the control group (Fig 2C–2D). Furthermore, the level of hydroxyproline was significantly elevated in AAV-shTTP-treated mice following *S. japonicum* infection (Fig 2F). Consistently, TTP knockdown led to the upregulation of α-SMA and collagen type 1 (COL1A1), the most abundant extracellular matrix (ECM) protein in the fibrotic hepatic tissues (Fig 2C). Furthermore, the hepatic mRNA levels of fibrogenic genes (*Acta2*, *Col1a1*, and *Col3a1*) were upregulated in TTP-knockdown mice of following *S. japonicum* infection (Fig 2G). Given that TTP is reported to negatively regulate CCL2, IL-6, and IL-8 expression by binding to AREs in their 3'UTRs [8,9], we measured the mRNA expression levels of these cytokines in TTP-knockdown mice following *S. japonicum* infection. Our results showed that TTP knockdown significantly upregulated the mRNA expression of *Il6*, *Il8*, and *Ccl2* in the *S. japonicum*-induced murine model of liver fibrosis (S3A Fig). Additionally, we observed that TTP knockdown significantly increased the mRNA expression levels of key Th1 (IFN-γ, TNF-α), Th2 (IL-4, IL-13), and Th17 (IL-17A) cytokines in this model (S3B–S3D Fig).

## Ectopic TTP attenuates S. japonicum-induced liver fibrosis *in vivo*

Next, we investigated whether ectopic TTP expression alleviates liver fibrosis *in vivo*. AAV8 vectors were used to deliver TTP to *S. japonicum*-infected mice (Fig 3A). AAV8-TTP-mediated TTP overexpression was confirmed in both non-infected

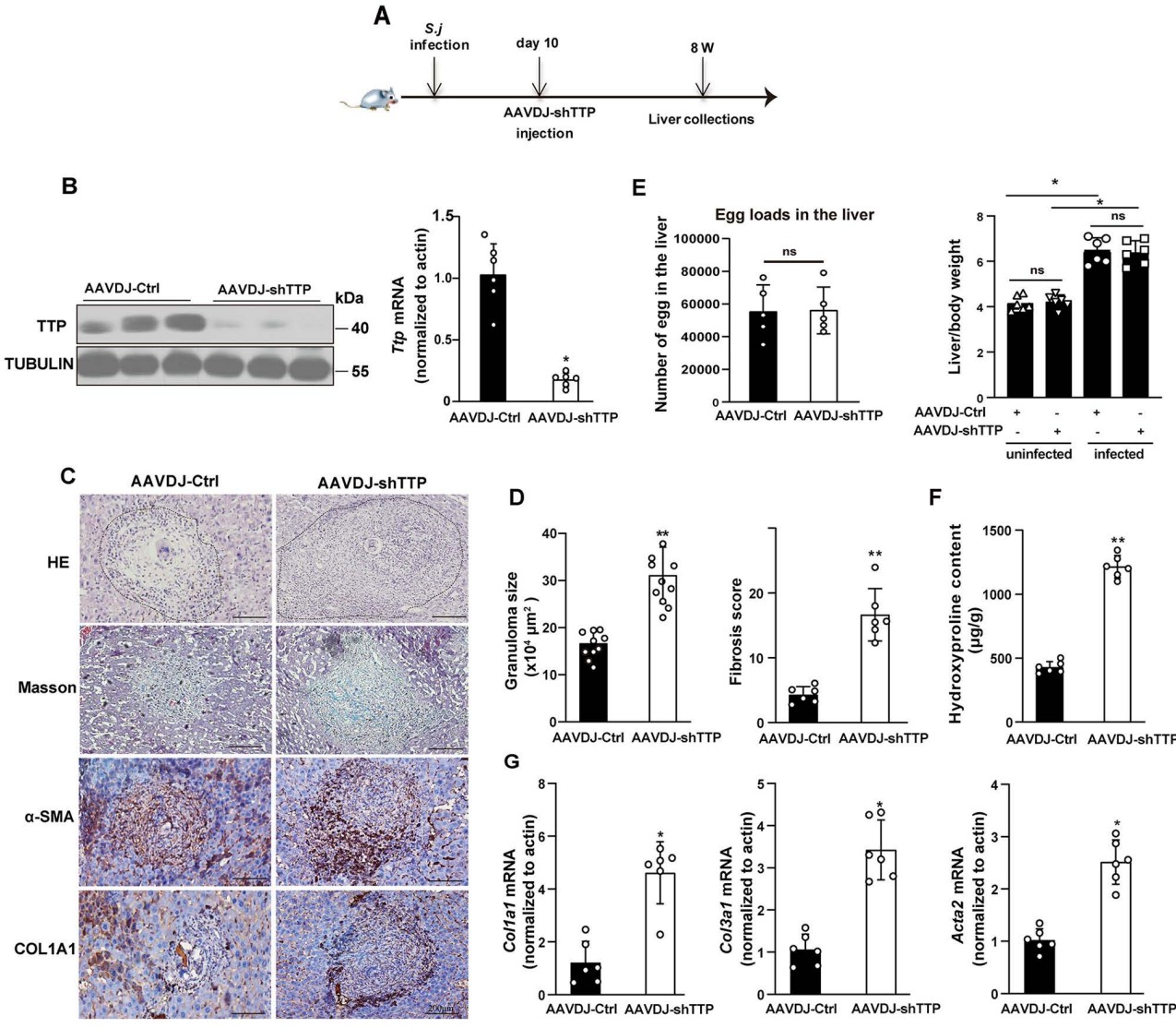

**Fig 2. TTP knockdown exacerbates liver fibrosis in mice with *S. japonicum* infection. (A)** Time schedule for parasite infection experiment. Mice were first infected with *S. japonicum*, and 10 days later, they were injected with AAVDJ-shTTP. The livers of the mice were collected at 8 weeks post-infection for subsequent analysis. **(B)** Western blot (right) and real-time PCR (left) analyses were performed to assess TTP expression in liver tissues of mice, following TTP knockdown mediated by AAV vectors (AAVDJ-shTTP) following *S. japonicum* infection. **(C)** H&E staining, Masson's trichrome staining, α-SMA staining, and COL1A1 staining of liver sections from the indicated groups (AAVDJ-shCtrl/infected and AAVDJ-shTTP/infected) (Scale bar: 200 μm). **(D)** Granuloma size (H&E staining) and fibrosis score (Masson's trichrome staining) were quantified via ImageJ software in multiple randomly selected fields across distinct liver sections. **(E)** The egg loads and the ratio of liver to body weight in the livers of AAVDJ-shCtrl and AAVDJ-shTTP-injected mice infected with *S. japonicum*. **(F)** The content of hydroxyproline in the liver was detected. **(G)** Real-time PCR analysis of *Acta2*, *Col1a1*, and *Col3a1* expression level in liver tissues. Data are presented as mean ± SD of 6 mice per group and are representative of three independent experiments. Statistical analyses were performed using 2-tailed Student's t-test. *$P < 0.05$ versus Ctrl. **$P < 0.01$ versus Ctrl. ns, no significant difference versus Ctrl.

and *S. japonicum*-infected mice (Figs 3B and S4A). H&E and Masson's trichrome staining confirmed that AAV8-Ctrl had no impact on *S. japonicum*-induced liver fibrosis relative to PBS control group, eliminating potential vector-related off-target effects (S4B–S4E Fig). Notably, ectopic TTP expression reduced the areas of egg granuloma in the livers of *S. japonicum*-infected mice (Fig 3C–3D). No significant differences were observed in liver egg loads or liver weight ratios

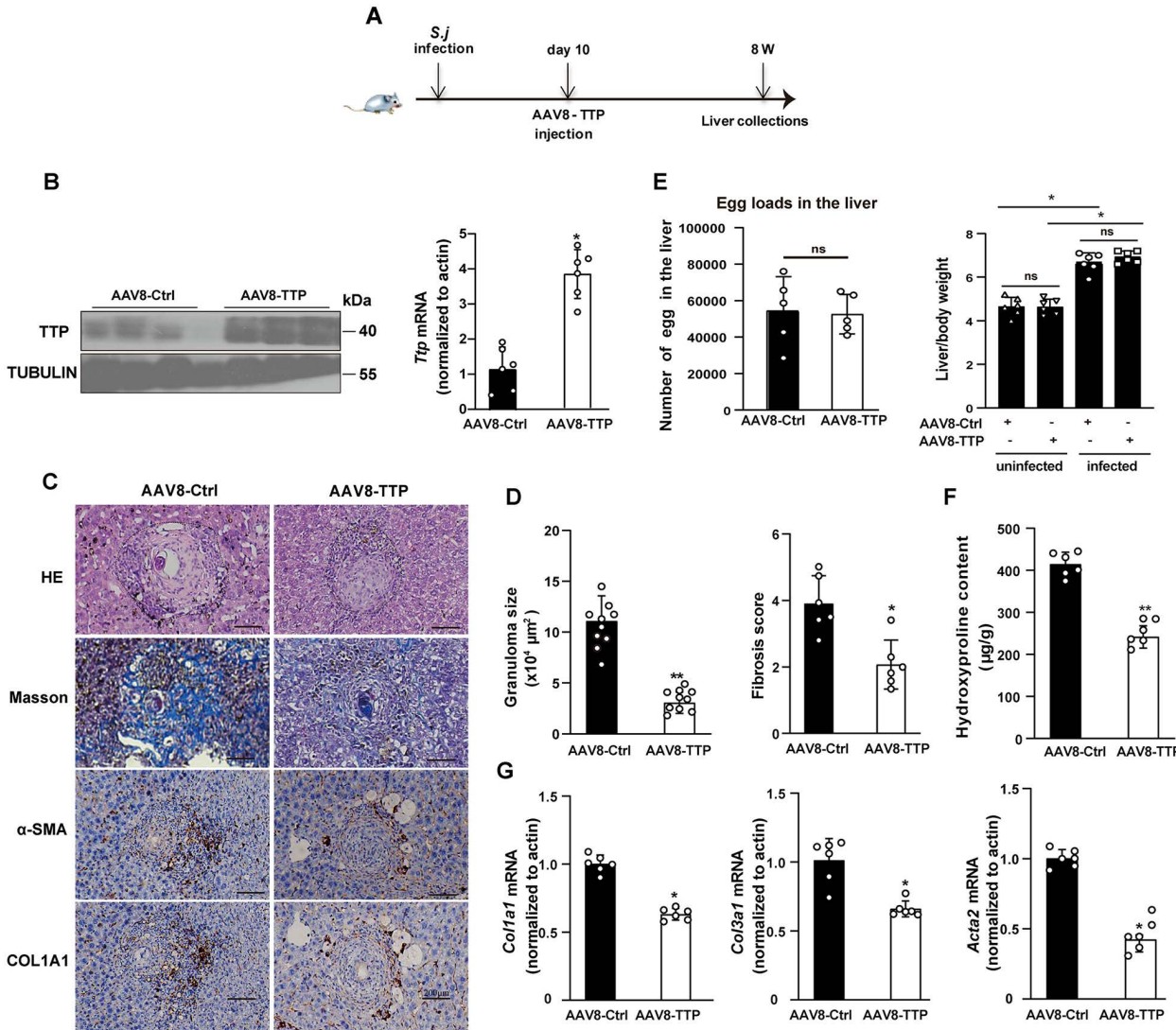

**Fig 3. TTP overexpression attenuates liver fibrosis in mice with *S. japonicum* infection. (A)** Time schedule for parasite infection experiment. Mice were first infected with *S. japonicum*, and 10 days later, they were injected with AAV8-TTP. The livers of the mice were collected at 8 weeks post-infection for subsequent analysis. **(B)** Western blot (right) and real-time PCR (left) analyses were performed to assess TTP expression in liver tissues of mice, following TTP over-expression mediated by AAV vectors (AAV8-TTP) following *S. japonicum* infection. **(C)** H&E staining, Masson's trichrome staining, α-SMA staining, and COL1A1 staining of liver sections from the indicated groups (AAV8-Ctrl/infected and AAV8-TTP/infected) (Scale bar: 200 μm). **(D)** Granuloma size (H&E staining) and fibrosis score (Masson's trichrome staining) were quantified via ImageJ software in multiple randomly selected fields across distinct liver sections. **(E)** The egg loads and the ratio of liver to body weight in the livers of AAV8-Ctrl and AAV8-TTP-injected mice infected with *S. japonicum*. **(F)** The content of hydroxyproline in the liver was detected. **(G)** Real-time PCR analysis of *Acta2*, *Col1a1*, and *Col3a1* expression level in liver tissues. Data are presented as mean±SD of 6 mice per group and are representative of three independent experiments. Statistical analyses were performed using 2-tailed Student's t-test. *$P<0.05$ versus Ctrl. **$P<0.01$ versus Ctrl. ns, no significant difference versus Ctrl.

between AAV-Ctrl and AAV8-TTP mice following *S. japonicum* infection (Fig 3E). Consistently, decreased hepatic fibrosis was observed in AAV8-TTP-treated mice following *S. japonicum* infection (Fig 3C–3D). TTP overexpression significantly attenuated the upregulation of a-SMA and Col1a1 in fibrotic hepatic tissues (Fig 3C). Moreover, the hydroxyproline level in AAV8-TTP-treated mice was reduced compared to that in control group after *S. japonicum* infection (Fig 3F). Furthermore,

hepatic mRNA levels of fibrogenic genes (*Acta2*, *Col1a1*, and *Col3a1*) were downregulated in the livers of *S. japonicum*-infected mice (Fig 3G).

## TTP negatively regulates the expression of TGF-β1 and MYB through m⁶A modification

To systematically elucidate the functional role of TTP in liver fibrosis, we re-analyzed our previously published RNA-Seq dataset from TTP-knockdown cells (GSE157581). The RNA-Seq data showed that multiple fibrosis-related genes, including *TGFB1*, *TIMP1*, *COL1A1*, *COL4A1*, and *MYB* showed a tendency to increase in TTP-knockdown cells (S5A Fig). The upregulation of TGF-β1, TIMP1, COL1A1, COL4A1, and MYB was validated in TTP-knockdown hepatocytes and LX-2 cells by real-time PCR (S5B Fig). Conversely, overexpression of TTP significantly reduced the mRNA expression of *TGFB1*, *TIMP1*, *COL1A1*, *COL4A1*, and *MYB* in both hepatocytes and LX-2 cells (S5C Fig).

Our previous study demonstrated that TTP regulates the mRNA stability of CCL2 and CCL5 via m⁶A RNA methylation [14]. To identify TTP-regulated genes via m⁶A modification, we profiled the m⁶A methylomes of TTP-knockdown and control cells using methylated RNA-immunoprecipitation sequencing (MeRIP-seq). MeRIP-seq analysis identified 33 upregulated and 81 downregulated m⁶A peaks, corresponding to 31 and 80 genes, respectively (Fig 4A). We further integrated MeRIP-seq and RNA-seq data in our analysis. Notably, we observed the downregulation of m⁶A peaks in *TGFB1* and *MYB* transcripts in TTP-knockdown cells (Figs 4B and S6A). Classic m⁶A motifs containing the UGGAC consensus sequences were identified in the 5′UTR of *TGFB1* and the coding sequences (CDSs) of *MYB* (Figs 4C and S6A).

To determine the role of m⁶A modifications in regulating TGF-β1 and MYB expression, hepatocytes were treated with 3-deazaadenosine (3-DAA), a global methylation inhibitor that suppresses m⁶A modification. 3-DAA treatment effectively upregulated the expression of TGF-β1 and MYB and markedly increased the mRNA half-lives of TGF-β1 and MYB (S6B–S6C Fig). We then generated two sets of luciferase reporter constructs: pMIR-TGF-β1–5′UTR and pMIR-MYB-CDS, in which luciferase cDNA was fused to the TGF-β1–5′UTR and MYB-CDS harboring wild-type m⁶A methylation sites, respectively; and their mutant counterparts (pMIR-TGF-β1–5′UTR mut and pMIR-MYB-CDS mut) (Fig 4D). As expected, TTP knockdown significantly increased the luciferase activities of pMIR-TGF-β1–5′UTR and pMIR-MYB-CDS (Fig 4E). In contrast, TTP overexpression reduced the luciferase activities of these constructs (S6D Fig). In cells transfected with the mutant constructs (pMIR-TGF-β1–5′UTR mut and pMIR-MYB-CDS mut), TTP expression had no significant effect on luciferase activities compared to the control group (Fig 4E). In addition, 3-DAA treatment significantly increased the luciferase activities of pMIR-TGFβ1–5′UTR and pMIR-MYB-CDS, but exerted no effect on that of the mutant constructs (S6E Fig). To further confirm whether TTP knockdown regulates m⁶A modification at the identified sites within the TGF-β1–5′UTR and MYB-CDS, we employed m⁶A immunoprecipitation-quantitative PCR (MeRIP-qPCR). Our data confirmed that m⁶A methylation was readily detectable at the m⁶A sites within the TGF-β1–5′UTR and MYB-CDS (Fig 4F). Furthermore, TTP knockdown reduced the m⁶A levels at these specific sites compared to the control (Fig 4F). Collectively, these results demonstrate that TTP negatively regulates the expression of TGF-β1 and MYB by promoting m⁶A modification at their respective m⁶A sites.

## TTP negatively regulates TGF-β1 and MYB expression through modulation of WTAP

Our previous study reported that TTP modulates m⁶A modification by regulating the expression of WTAP, METTL14, and YTHDF2 [14]. We thus investigated the potential involvement of these three molecules in the regulation of TGF-β1 and MYB expression. Overexpression of WTAP significantly decreased the luciferase activities of the TGF-β1–5′UTR and MYB-CDS reporter constructs (Fig 5A). In contrast, overexpression of METTL14 or YTHDF2 did not affect the luciferase activities of these two reporter constructs (S7A Fig). Moreover, WTAP knockdown significantly increased the luciferase activities of constructs containing the TGF-β1–5′UTR and MYB-CDS in both hepatocytes and LX-2 cells (Figs 5B and S7B). However, this effect was abolished in constructs with mutated m⁶A sites (Figs 5B and S7B). Knockdown of WTAP also upregulated the mRNA expression of TGF-β1 and MYB in hepatic cells (Fig 5C). We next investigated whether WTAP

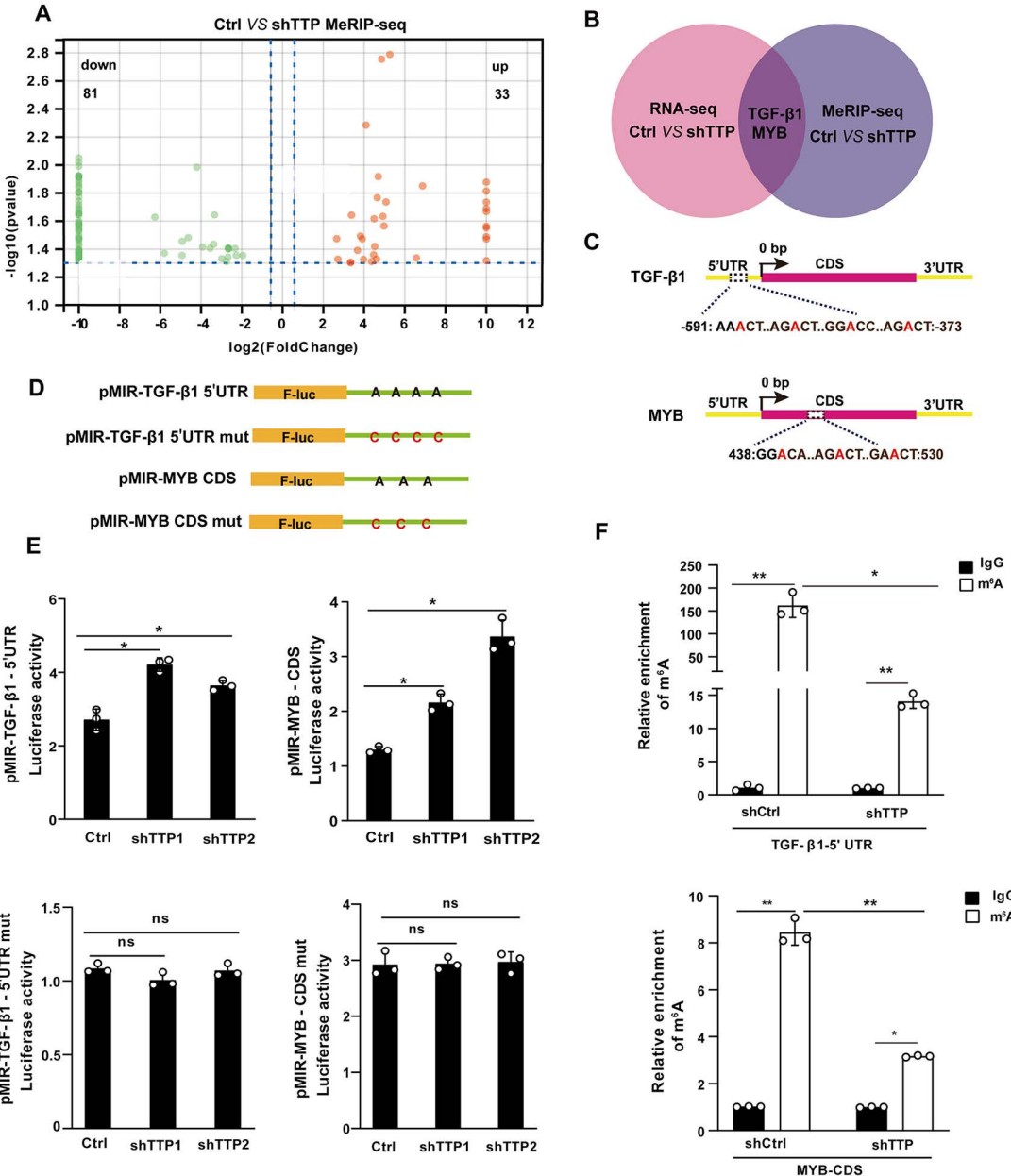

**Fig 4. TTP negatively regulates the stabilization of TGF-β1 and MYB mRNAs through m⁶A modification. (A)** Volcano plots showing up or down-regulated genes in TTP-knockdown hepatic cells versus Ctrl, as assessed by meRIP-seq. **(B)** A Venn diagram was generated using gene sets enriched in transcripts that were substantially altered after TTP knockdown (RNA-seq) and those enriched in m⁶A-modified transcripts (meRIP-seq). **(C)** Schematic representation of the position of m⁶A motifs within the *TGFB1* and *MYB* transcripts. **(D)** Wild-type or mutated m⁶A consensus sequences (A-to-C mutation) of pMIR-TGF-β1-5'UTR and pMIR-MYB-CDS were fused with Renilla luciferase reporter in pMIR-GLO vector. **(E)** Stable TTP-knockdown hepatic cells were transfected with pMIR-TGF-β1 5'UTR, pMIR-MYB-CDS, pMIR-TGF-β1-5'UTR mut, or pMIR-MYB-CDS mut for 24 h, followed by luciferase analysis. **(F)** MeRIP-qPCR analysis of m⁶A enrichment on TGF-β1 and MYB in TTP knockdown cells. Data are presented as mean ± SD of three independent experiments. Statistical analyses were performed using one-way ANOVA and 2-tailed Student's t-test. *$P < 0.05$ versus Ctrl. **$P < 0.01$ versus Ctrl. ns, no significant difference versus Ctrl.

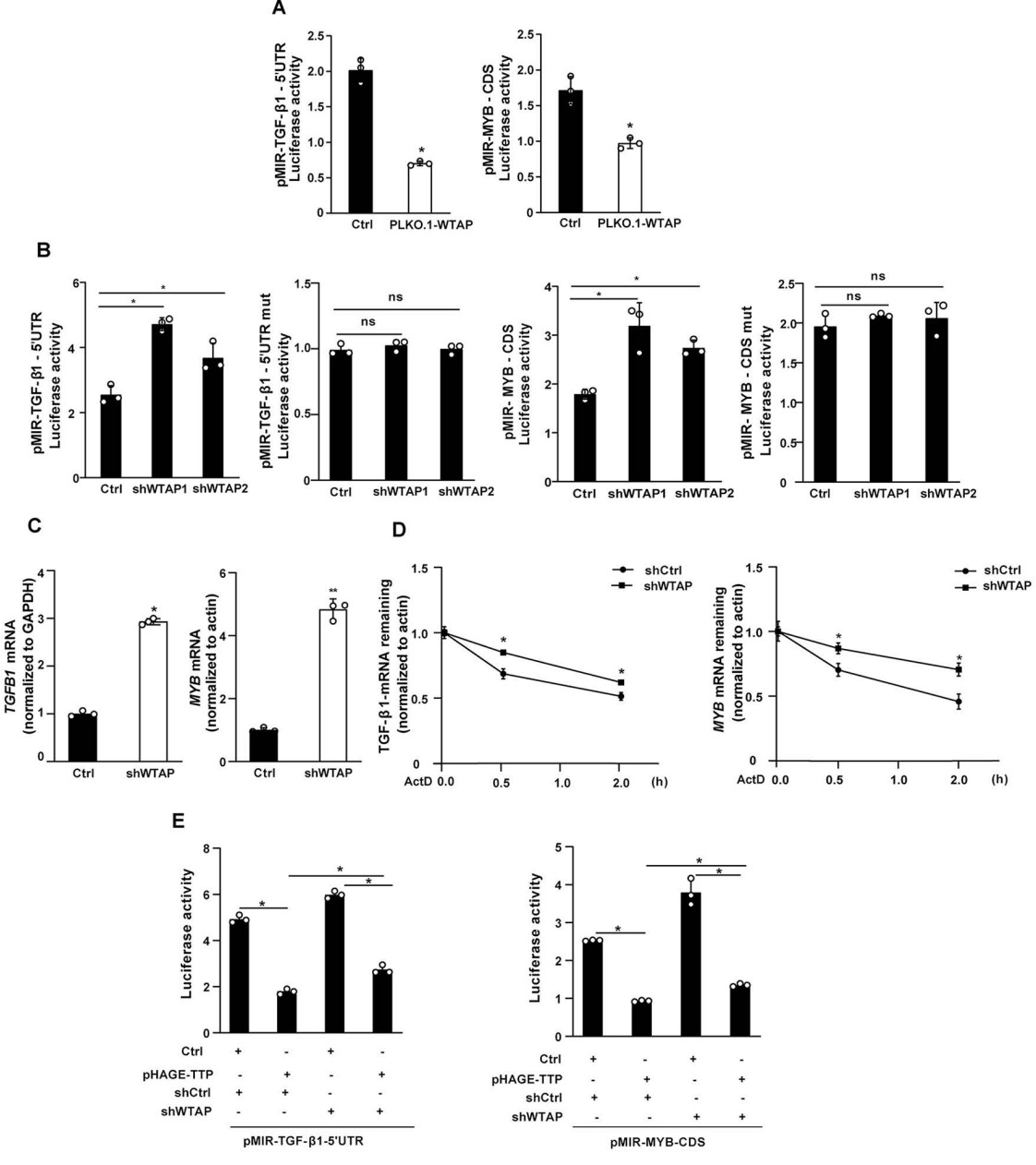

**Fig 5. WTAP-mediated m⁶A methylation regulates TGF-β1 and MYB mRNA stability.** **(A)** Overexpression of WTAP in hepatic cells were transfected with pMIR-TGF-β1-5'UTR or pMIR-MYB-CDS, followed by luciferase analysis. **(B)** Stable WTAP-knockdown hepatic cells were transfected with pMIR-TGF-β1-5'UTR, pMIR-MYB-CDS, pMIR-TGF-β1-5'UTR mut or pMIR-MYB-CDS mut for 24 h, followed by luciferase analysis. **(C)** The mRNA level of *TGFB1* and *MYB* was analyzed in stable WTAP knockdown hepatic cells cells by real-time PCR. **(D)** The stability of *TGFB1* and *MYB* mRNA in stable WTAP knockdown hepatic cells. **(E)** TTP overexpression hepatic cells were co-transfected with shWTAP and the luciferase pMIR-TGF-β1-5'UTR or pMIR-MYB-CDS, followed by luciferase analysis. Data are presented as mean ± SD of three independent experiments. Statistical analyses were performed using 2-tailed Student's t-test, one-way ANOVA, and two-way ANOVA. *$P < 0.05$ versus Ctrl. **$P < 0.01$ versus Ctrl. ns, no significant difference versus Ctrl.

represses TGF-β1 and MYB expression by modulating the stability of their respective mRNAs. Indeed, WTAP knockdown significantly increased the mRNA half-lives of both TGF-β1 and MYB (Fig 5D).

Given that TTP regulates TGF-β1 and MYB expression via their m6A methylation sites and WTAP is involved in modulating their expression, we hypothesized that TTP influences m6A-mediated post-transcriptional regulation of TGF-β1 and MYB by regulating WTAP expression. To test this hypothesis, cells were co-transfected with pHAGE-TTP (for TTP overexpression) and WTAP-specific shRNAs. As expected, TTP overexpression reduced the luciferase activities of pMIR-TGFβ1–5′UTR and pMIR-MYB-CDS constructs (Fig 5E). Furthermore, WTAP shRNAs abrogated the TTP overexpression-mediated increase in luciferase activities (Fig 5E). Collectively, these findings indicate that TTP negatively regulates TGF-β1 and MYB expression via modulation of WTAP expression.

## TTP interacts with SMAD2/3

To further elucidate the mechanism by which TTP transcriptionally regulates m6A modification-associated genes, we performed immunoprecipitation using an anti-FLAG antibody in HEK293T cells transfected with pHAGE-TTP plasmids (for TTP overexpression). Then, we performed liquid chromatography-tandem mass spectrometry (LC-MS/MS) to identify the potential binding partners of TTP (Fig 6A). A total of 198 proteins (≥10high-quality PSMs: peptide spectrum matches) were identified (PRIDE database: PXD056357, S3 Table). Metascape enrichment analysis revealed that these proteins were primarily involved in metabolism of RNA, regulation of mRNA metabolic process, mRNA processing, regulation of RNA splicing, chromosome organization, ribonucleoprotein complex biogenesis, regulation of chromosome organization, positive regulation of protein localization to nucleus, CDC5L complex, signaling by TGF-β family members (Fig 6B). Notably, both mass spectrometry data and our previous coimmunoprecipitation (Co-IP) results indicated that TTP does not interact with WTAP [14]. Interestingly, SMAD2 and SMAD3 were identified as TTP-interacting proteins (Fig 6C). SMAD2 and SMAD3 are well-established as core transcriptional regulators of the TGF-β signaling pathway, a key pathway critically involved in the pathogenesis of *S. japonicum*-induced liver fibrosis [17,18]. The interactions between TTP and SMAD2/3 were confirmed by using Co-IP assays in both HEK293T and HepG2 cells (Figs 6D and S7C). To identify the critical TTP domains responsible for SMAD2/3 binding, HEK293T cells were co-transfected with FLAG-tagged serially truncated TTP and HA-tagged SMAD2/3 constructs (Fig 6E). The N-Zn (1–200 aa) and Zn-C (200–326 aa) domains of TTP were found to be pivotal for mediating its interaction with SMAD2/3 (Fig 6F). Moreover, we aimed to identify the specific segments of SMAD2/3 that are capable of interacting with TTP. Domain truncation combined with Co-IP assays revealed that the MH1 (SMAD2: 1–172 aa, SMAD3: 1–132 aa) and MH2 (SMAD2: 266–456 aa, SMAD3: 224–414 aa) domains of SMAD2/3 were required for TTP binding (Fig 6G–6H). Overall, these in vitro binding assays demonstrate that TTP interacts with both SMAD2 and SMAD3.

## TTP promotes WTAP and METTL14 expression via interaction SMAD2/3

Given that TTP transcriptionally regulates WTAP, METTL14, and YTHDF2 expression and physically interacts with SMAD2/3, we hypothesized that TTP promotes the expression of these m6A-related factors through its interaction with SMAD2/3. To investigate the regulatory effects of SMAD2/3 on the transcriptional regulation of WTAP, METTL14, and YTHDF2, cells were co-transfected with the luciferase reporter plasmids containing *WTAP*, *METTL14*, or *YTHDF2* promoter regions and SMAD2/3 overexpression vectors. Overexpression of SMAD2/3 significantly enhanced the luciferase activities of the *WTAP* and *METTL14* promoter reporter constructs, as measured by luciferase assay in HEK293T cells (Fig 7A). By contrast, overexpression of SMAD2/3 had no significant effect on the luciferase activity of *YTHDF2* promoter (S7D Fig). Conversely, the knockdown of SMAD2/3 effectively repressed the luciferase activities of *WTAP* and *METTL14* promoter regions (Fig 7A).

Given that the N-Zn or Zn-C domains of TTP interact with SMAD2/3, we next investigated the effect of these TTP domains on the promoter activities of *WTAP* and *METTL14*. Overexpression of TTP-N-Zn or TTP-Zn-C enhanced the

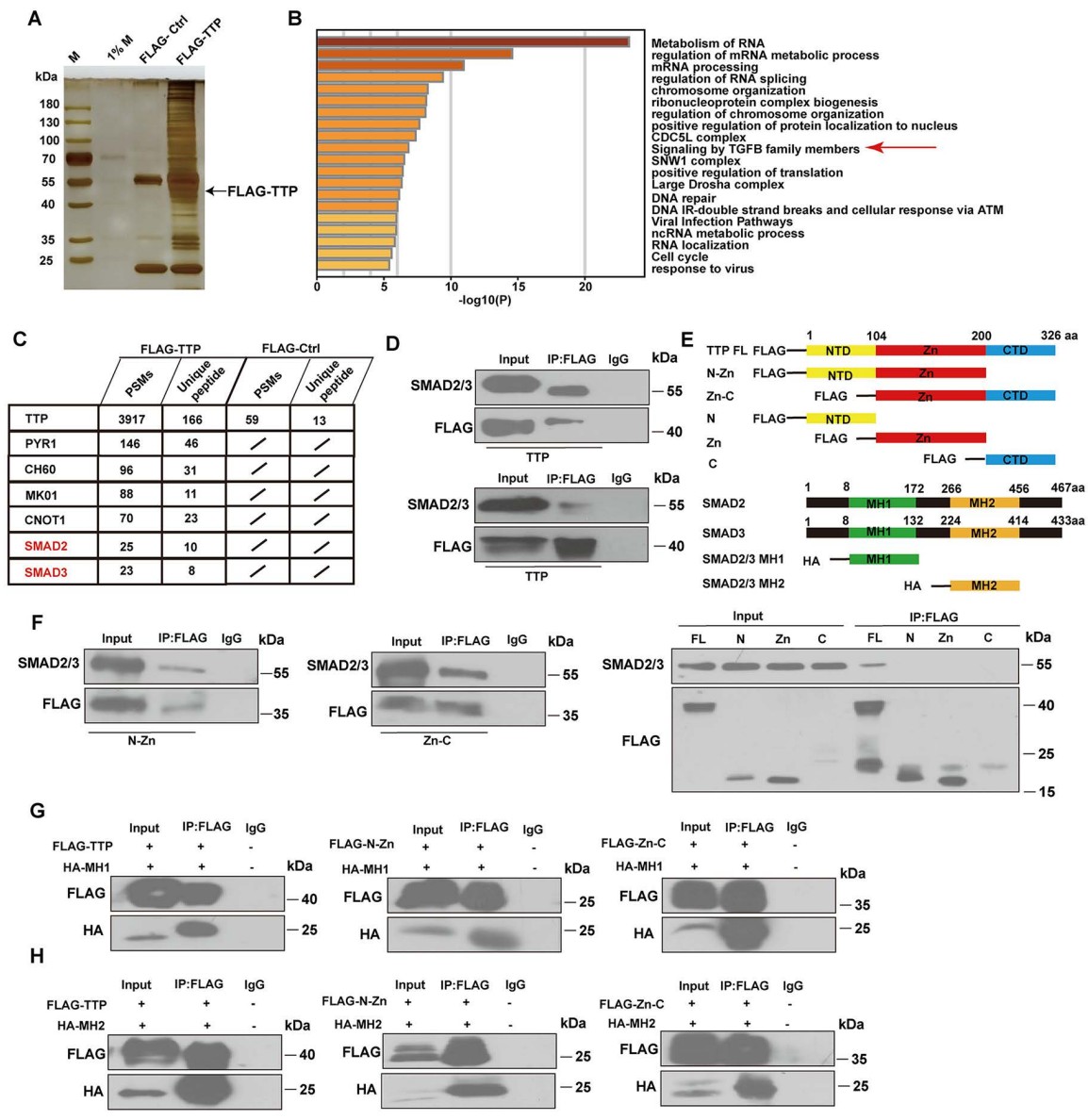

**Fig 6. TTP interacts with SMAD2/3. (A)** Silver staining was used to visualize interacting proteins with TTP, which were enriched using a FLAG antibody. The arrowhead indicates the predicted size of TTP. **(B)** Gene ontology (GO) based on immunoprecipitation results showing signaling pathway of the TTP interactome. **(C)** TTP-interacting proteins enriched over IgG. Data from mass spectrometry analysis are shown in the table. Peptide-spectrum match counts (#PSM). SMAD2 and SMAD3 are labeled in red. **(D)** Exogenous FLAG-tagged TTP and SMAD2/3 were immunoprecipitated in HepG2 and HEK293T cells. **(E)** Schematic diagrams of TTP (top) and SMAD2/3 (bottom) truncations. **(F)** TTP-N-Zn, TTP-Zn-C, TTP-C, TTP-Zn or TTP-C and SMAD2/3 were immunoprecipitated in HEK293T cells. **(G)** Interactions between TTP-N-Zn or TTP-Zn-C and SMAD2/3 MH1 in HEK293T cells were examined by Co-IP. **(H)** Interactions between TTP-N-Zn or TTP-Zn-C and SMAD2/3 MH2 in HEK293T cells were examined by Co-IP.

luciferase activities of *WTAP* and *METTL14* promoter reporter constructs (Fig 7B). In contrast, overexpression of the individual N, Zn or C single domains did not affect the luciferase activities of *WTAP* and *METTL14* promoter reporter constructs (S7E Fig). These data further confirm that the N-Zn and Zn-C domains of TTP are critical for mediating its interaction with SMAD2/3.

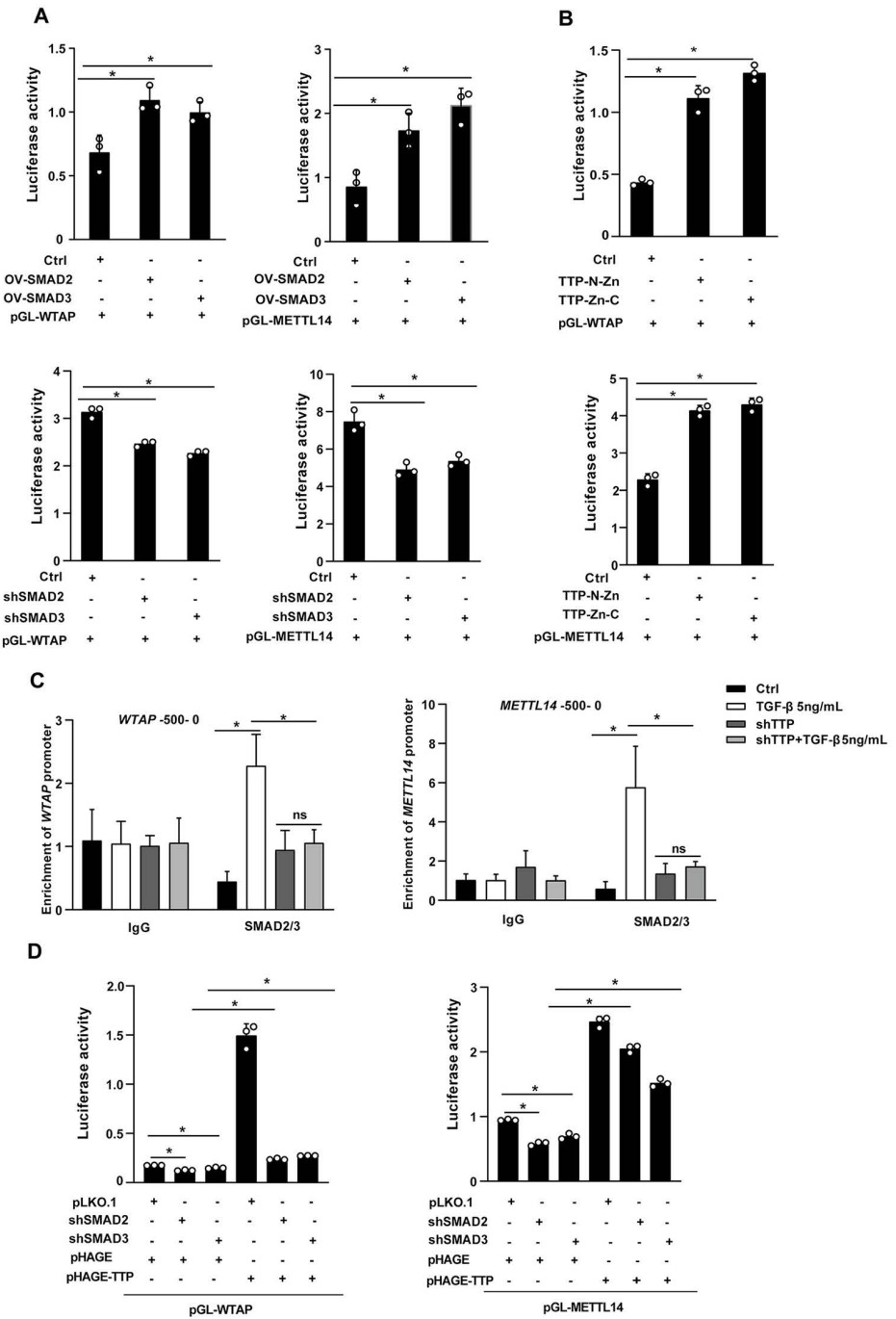

**Fig 7. TTP promotes WTAP and METTL14 expression via interaction SMAD2/3. (A)** HEK293T cells with stable SMAD2/3 knockdown or overexpression were transfected with the promoters of WTAP or METTL14, followed by luciferase reporter assay 24 h later. **(B)** HEK293T cells were co-transfected with the promoters of *WTAP* or *METTL14*, along with TTP-N-Zn or TTP-Zn-C, followed by luciferase analysis. **(C)** The recruitment of SMAD2/3 to the promoter regions of *WTAP* and *METTL14* was detected by ChIP-PCR analysis. **(D)** TTP overexpression HEK293T cells were co-transfected with shMAD2, shSMAD3, and the promoters of *WTAP* and *METTL14*, followed by luciferase analysis. Data are presented as mean ± SD of three independent experiments. Statistical analyses were performed using one-way ANOVA, two-way ANOVA, and 2-tailed Student's t-test. *$P < 0.05$ versus Ctrl. ns, no significant difference versus Ctrl.

Considering that SMAD2 and SMAD3 function as key transcriptional activators in the TGF-β signaling network, we performed real-time PCR to quantify the mRNA expression of WTAP and METTL14 in cells following TGF-β stimulation. TGF-β treatment significantly increased the mRNA expression of WTAP and METTL14 (S7F Fig). To further explore whether TTP promotes *WTAP* and *METTL14* transcription via interacting with SMAD2/3, we performed ChIP-PCR assays using primers spanning the promoter regions (−500–0 bp) of *WTAP* and *METTL14*. ChIP-PCR analysis revealed that TGF-β stimulation markedly increased the enrichment of SMAD2/3 on the well-characterized promoter regions of WT*AP* and *METTL14* (Fig 7C). TTP knockdown substantially abrogated the TGF-β-induced enrichment of SMAD2/3 on the *WTAP* and *METTL14* promoters (Fig 7C). Furthermore, luciferase reporter assays were performed in stable SMAD2 and SMAD3 knockdown HEK293T cells, which were co-transfected with TTP overexpression plasmids and luciferase reporter vectors carrying the *WTAP* or *METTL14* promoters. SMAD2 and SMAD3 knockdown abolished the upregulation of luciferase activity induced by TTP overexpression (Fig 7D). Therefore, these results suggest that TTP promotes the transcription of *WTAP* and *METTL14* through its interaction with SMAD2/3.

## TTP modulates HSC activation by regulating TGF-β1 expression

Given that TTP negatively regulates TGF-β1 expression and is significantly upregulated in hepatocytes from murine liver fibrosis models, we hypothesized that TTP modulates HSC activation via regulating TGF-β1 expression. To test this hypothesis, we co-cultured HSCs with hepatocytes stably transfected with shTTP in the presence or absence of a TGF-β1-neutralizing antibody. TTP knockdown increased the mRNA expression of *SMAD7*, *COL1A1*, and *ACTA2* in LX-2 cells (S8A Fig). Additionally, the TGF-β1-neutralizing antibody significantly suppressed the TTP knockdown-induced upregulation of *SMAD7*, *COL1A1*, and *ACTA2* expression in LX-2 cells (S8A Fig). To evaluate whether TTP induces HSC activation by regulating WTAP expression, stable WTAP-knockdown hepatic cells were transfected with pHAGE-TTP or empty control vectors. These transfected hepatic cells were then co-cultured with HSCs. Real-time PCR analysis revealed that WTAP knockdown significantly increased the expression of *SMAD7*, *COL1A1*, and *ACTA2* in LX-2 cells (S8B Fig). Notably, TTP overexpression significantly reversed the WTAP knockdown-induced upregulation of *SMAD7*, *COL1A1*, and *ACTA2* in LX-2 cells (S8B Fig).

## m⁶A RNA methylation contributes to TTP-mediated protection against liver fibrosis

Considering that TTP promotes WTAP and METTL14 expression, we hypothesized that m⁶A RNA methylation may contribute to the pathogenesis of liver fibrosis. To test this hypothesis, we first evaluated the global m⁶A levels in liver tissues from *S. japonicum*-infected mice using dot blot assay. Global m⁶A levels significantly higher in liver tissues from mice with *S. japonicum*-induced liver fibrosis than in control mice (S9A Fig). Moreover, immunohistochemical analysis revealed that the protein levels of WTAP, METTL14, and YTHDF2 were increased in fibrotic liver tissues (S9B Fig). Additionally, we measured the mRNA expression of TGF-β1, MYB, WTAP, METTL14 and YTHDF2 in mouse primary hepatocytes and LX-2 cells stimulated with SEA. Our data demonstrate that SEA stimulation significantly upregulated the expression of TGF-β1, MYB, WTAP and METTL14 in both cell types (S9C Fig). By contrast, no significant change in YTHDF2 mRNA expression was observed in either cell type following SEA treatment (S9C Fig).

To investigate whether TTP-mediated protection against *S. japonicum*-induced liver fibrosis is linked to the regulation of m⁶A RNA methylation *in vivo,* 3-DAA, an inhibitor of global methylation, was administered via intraperitoneal injection to AAV8-TTP-infected mice and control mice at 42 days post *S. japonicum* infection (Fig 8A). It is well established that schistosome eggs start depositing in the liver at approximately 42 days post-infection, triggering robust hepatic granuloma formation and HSC activation [19]. By this time point, liver fibrosis is fully established, rendering it an optimal window for evaluating the efficacy of 3-DAA. Additionally, in vivo toxicity assessments showed no signs of systemic toxicity in 3-DAA-treated mice, as evidenced by stable body weight, normal liver function (reflected by ALT/AST levels), and the absence of abnormal histopathological alterations (S9D–S9E Fig). Dot blot analysis confirmed that 3-DAA treatment not only

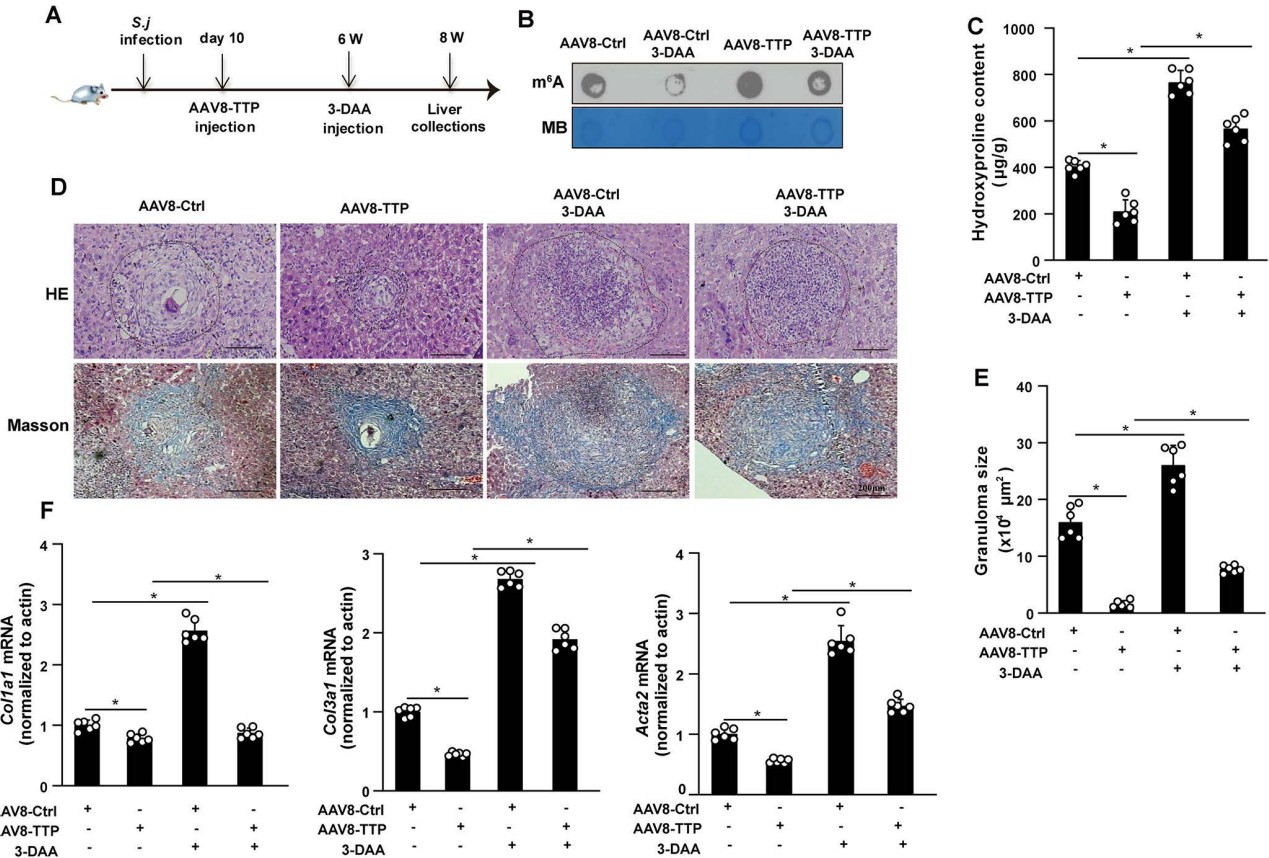

**Fig 8. m⁶A RNA methylation contributes to TTP-mediated protection against *S. japonicum*-induced liver fibrosis.** (A) Time schedule for parasite infection experiment. Mice were first infected with *S. japonicum*, and 10 days later, they were injected with AAV8-TTP. At 6 weeks post-infection, 3-DAA was administered via injection. The livers of the mice were collected at 8 weeks post-infection for subsequent analysis. (B) The m⁶A RNA modification was determined by Dot blot. (C) The content of hydroxyproline in the liver was detected. (D) H&E staining and Masson's trichrome staining of liver sections from the indicated groups (AAV8-Ctrl/infected, AAV8-TTP/infected, AAV8-Ctrl/infected+3-DAA, and AAV8-TTP/infected+3-DAA) (Scale bar: 200 µm). (E) Granuloma size (H&E staining) was quantified via ImageJ software in multiple randomly selected fields across distinct liver sections. (F) Real-time PCR analysis of *Acta2*, *Col1a1*, and *Col3a1* expression level in liver tissues. Data are presented as mean ± SD of 6 mice per group and are representative of three independent experiments. Statistical analyses were performed using two-way ANOVA. *$P < 0.05$ versus Ctrl.

significantly reduced global m⁶A levels in mouse liver tissues, but also abolished the TTP overexpression-induced m⁶A elevation in *S. japonicum*-infected livers (Fig 8B). Moreover, 3-DAA treatment significantly decreased hepatic hydroxyproline levels in *S. japonicum*-infected mice overexpressing TTP (Fig 8C). Importantly, H&E and Masson's trichrome staining revealed that 3-DAA treatment abrogated the hepatic protective effect conferred by TTP overexpression in *S. japonicum*-infected mice (Fig 8D–8E). Moreover, upregulation of several fibrosis-related factors, including *Col1a1*, *Col3a1*, and *Acta2*, was detected in *S. japonicum*-infected mice treated with 3-DAA, compared with vehicle-treated control group (Fig 8F). Collectively, these findings indicate that m⁶A RNA methylation contributes to TTP-mediated protection against liver fibrosis.

## Discussion

This study first demonstrates that dysregulated TTP-guided m⁶A RNA hyper- modifications serves as an essential regulator of HSC activation in *S. japonicum*-induced liver fibrosis. Elevated TTP expression promotes *WTAP* transcription via

interaction with SMAD2/3. As a key m$^6$A RNA modification methyltransferase "writers", WTAP downregulates TGF-β1 in an m$^6$A-dependent manner, thereby ultimately modulating HSC activation in *S. japonicum*-induced liver fibrosis (Fig 9).

TTP is significantly upregulated in sepsis- and *Echinococcus multilocularis*-induced liver injury [20,21], and our prior work confirmed elevated TTP expression in APAP- and CCl4-induced murine acute liver injury models [14]. However, TTP expression in liver fibrosis is controversial: it is downregulated in chronic hepatitis B-induced fibrosis [22], yet upregulated in high-fat diet-induced hepatic steatosis and CCl4-induced liver fibrosis [11]. In line with previous reports, we observed the upregulation of TTP in in a murine model of *S. japonicum*-induced liver fibrosis. Interestingly, we further found that TTP expression was progressively upregulated in parallel with the increasing severity of liver fibrosis. The expression of TTP is tightly regulated by multiple intracellular signaling pathways, including the p38MAPK and NF-κB signaling pathways [23,24]. Moreover, robust activation of both pathways is well-documented in liver fibrosis [25], which may underlie the upregulated TTP expression observed in this pathological context.

TTP, recognized as a transcriptional co-regulator, can interacts with various transcription factors, including NF-κB, thereby regulating gene expression at the transcriptional level [26]. Our previous study demonstrated that TTP promotes *WTAP*, *METTL14*, and *YTHDF2* transcription, thereby enhancing global m$^6$A RNA methylation, though the specific regulatory mechanisms remain to be fully elucidated [14]. Using mass spectrometry and co-immunoprecipitation assays, we found that TTP interacts with SMAD2/3 via its N- and C-terminal zinc-finger domains. SMAD2 and SMAD3 are core transcription factors in the TGF-β signaling pathway [27], and our findings further confirmed that the N- and C-terminal zinc-finger domains of TTP bind to the MH1 and MH2 domains of SMAD2/3. Notably, SMAD2 forms a complex with SMAD3 to modulate the transcription of target gene. Specifically, both SMAD2 and SMAD3 contain highly conserved MH1 and MH2 domains, with the MH1 domain playing a pivotal role in TGF-β signaling transduction by directly binding to DNA at the SMAD-binding element (SBE) within the promoter regions of target genes [28]. We also found that TGF-β stimulation induces the transcription of *WTAP* and *METTL14*. Our ChIP-PCR results demonstrated TTP knockdown abrogates the TGF-β-induced increase in SMAD2/3 binding to the promoters of *WTAP* and *METTL14*. Collectively, these findings support that TTP collaborates with SMAD2/3 to promote WTAP and METTL14 expression, which provides important insights into TTP functioning as a transcriptional co-regulator. Notably, our current data cannot definitively distinguish between direct

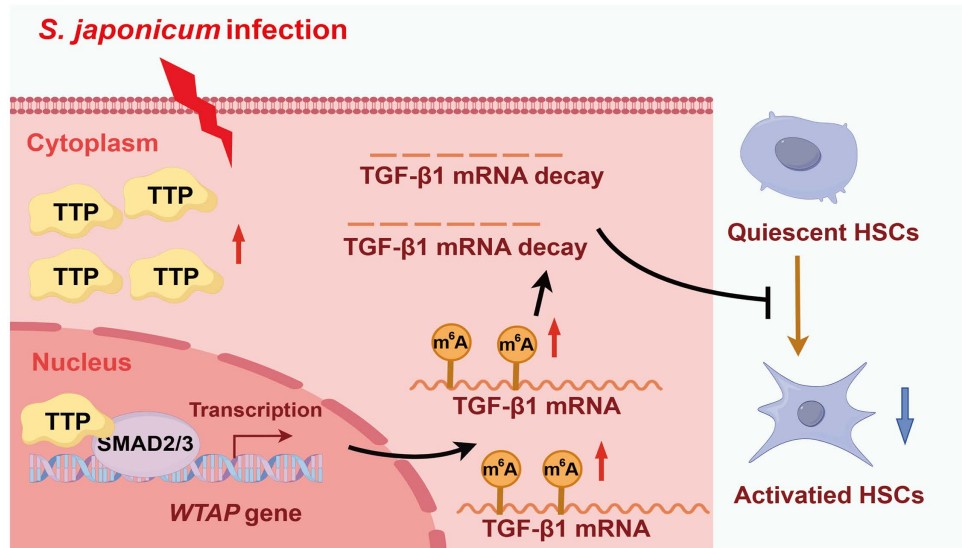

**Fig 9. TTP alleviates schistosomiasis-induced liver fibrosis through m$^6$A-dependent modulation of TGF-β1 mRNA stability.** TTP, which is significantly upregulated in *S. japonicum*-induced liver fibrosis, exerts a protective effect against liver fibrosis progression via interaction with SMAD2/3 to promote *WTAP* transcription; moreover, inhibition of TGF-β1 m$^6$A mRNA methylation effectively alleviates HSC activation.

and indirect transcriptional regulatory effects of TTP on these target genes, and the precise underlying mechanisms warrant further investigation. Additionally, SMAD2/3 overexpression did not alter *YTHDF2* promoter luciferase activity, implying the TTP-SMAD2/3 complex-mediated gene regulatory mechanism characterized in our study does not apply to YTHDF2.

HSC activation is widely recognized as the central event driving hepatic fibrosis progression [29]. TGF-β, a key pro-fibrotic cytokine secreted by hepatocytes and macrophages, initiates the fibrotic signaling cascade [30]. Additionally, MYB is well-documented to induce the expression of α-SMA during stellate cell activation [31]. Notably, integrated analysis of RNA-seq and MeRIP-seq data from TTP-knockdown cells revealed that TTP negatively regulates TGF-β1 and MYB expression by reducing their mRNA stability in an m$^6$A-dependent manner. Interestingly, our luciferase reporter assay results indicate that YTHDF2 is unlikely to act as a key "reader" protein for these two genes. Our data further suggest the existence of an alternative m$^6$A recognition mechanism governing these two genes' mRNA stability, with their specific cognate "reader" proteins yet to be identified in future studies. Combined with prior findings, our data further demonstrate that TTP enhances m$^6$A modification of these target mRNAs. Importantly, TTP exerts a protective role against liver fibrosis via negative regulation of TGF-β1 and MYB expression in both hepatocytes and HSCs. Although TTP is expressed in hepatocytes, its regulatory effect on HSC activation is mediated through paracrine TGF-β1 signaling. Collectively, these findings underscore that TTP acts as a key regulatory factor in the pathogenesis of liver fibrosis by repressing the expression of TGF-β1 and MYB.

Recent studies have uncovered intricate connections between m$^6$A RNA methylation and human liver diseases [32]. Emerging evidence further highlights the dysregulation of m$^6$A RNA methylation in the pathogenesis of diverse liver disorders, including non-alcoholic fatty liver disease (NAFLD), HBV-associated hepatitis, HCV-associated hepatitis, and hepatocellular carcinoma (HCC) [33]. Our findings provide compelling evidence for significantly elevated m$^6$A RNA methylation levels in a murine model of schistosomiasis-induced liver fibrosis. Additionally, the expression of m$^6$A-related genes (*WTAP*, *METTL14*, and *YTHDF2*) was markedly upregulated in the same murine model. Furthermore, in 3-DAA-treated TTP-overexpressing mice with *S. japonicum*-induced liver fibrosis, we further confirmed that TTP overexpression attenuates *S. japonicum*-induced liver fibrosis by modulating m$^6$A RNA methylation.

Although TTP has been reported to negatively regulate CCL2, IL-6, and IL-8 expression by binding to AREs in their 3'UTRs [8,9], our study focused on elucidating the mechanism by which TTP modulates TGF-β1 mRNA stability via m$^6$A methylation through its interaction with SMAD2/3. TTP-mediated m$^6$A-dependent degradation of TGF-β1 mRNA thus unveils an additional regulatory layer in the pathogenesis of *S. japonicum*-induced liver fibrosis. As noted, the TTP alleviates schistosomiasis-induced liver fibrosis through m$^6$A-mediated regulation of TGF-β1 mRNA stability, and this mechanism is not restricted to schistosomiasis-induced fibrosis alone. These findings highlight the intricate crosstalk between TTP and m$^6$A methylation in the pathogenesis of *S. japonicum*-induced liver fibrosis, and uncover a promising therapeutic strategy for treating this disease by targeting TTP or m$^6$A methylation.

In summary, our study elucidates a pivotal role for TTP in the pathogenesis of *S. japonicum*-induced liver fibrosis, whereby it post-transcriptionally regulates TGF-β1 in an m$^6$A-dependent manner both *in vitro* and *in vivo*. Specifically, WTAP-mediated m$^6$A RNA methylation of *TGFB1* transcript represses HSC activation, thereby attenuating schistosomiasis-induced liver fibrosis. Furthermore, our data uncover a previously unreported mechanism whereby TTP modulates *WTAP* transcription via direct interaction with SMAD2/3. Collectively, these findings demonstrate that TTP-driven transcriptional regulation of *WTAP* constitutes a novel mechanism underlying HSC activation and highlights potential therapeutic targets for *S. japonicum*-induced liver fibrosis intervention.

## Materials and methods

### Ethics statement

All experimental mice were carried out in strict accordance with the recommendations in the Guide for the Care and Use of Laboratory Animals of the Ministry of Science and Technology of the People's Republic of China. The animal protocol was approved by Ethic Committee of Animal Facility, Wuhan University (Approval Number: WQ20210053).

## Establishment of the hepatic fibrosis models using *S. japonicum*

Six-week-old male BALB/c mice, weighing 18–20 g, were purchased from the Hubei Provincial Center for Disease Control and Prevention (Wuhan, China). *Oncomelania hupensis* snails were obtained from the Snail Ecological Station of Gongan County (Jingzhou, China). BALB/c mice were used to establish a *S. japonicum*-induced liver fibrosis model, via percutaneous infection with 16 *S. japonicum* cercariae. 3-deazaadenosine (3-DAA, APExBIO, USA, Cat. No. B6121), a methylation inhibitor, was intraperitoneally (i.p.) injected into mice at 1.5 µg/g body weight every two days, starting at 4 weeks postinfection. Tissues were harvested at 8 weeks post-infection, with seven total injections administered.

## Cell lines and reagents

The human hepatic cell lines HL7702, HepG2, and human embryonic kidney cell line HEK293T were obtained from the American Type Culture Collection (ATCC). The human HSC line LX-2 was kindly provided by Dr. Yong Wang (Nanjing Medical University) and was originally obtained from ATCC. Soluble egg antigen (SEA) was prepared from purified eggs from the livers of *S. japonicum*-infected mice, as previously described [34]. SEA was used at a final concentration of 30 µg/mL for cell lines or 60 µg/mL for primary cells [34].

## Isolation of primary hepatocytes and non-parenchymal liver cells

Primary hepatocytes and HSCs were isolated from experimental mice using a previously established protocol [35]. In brief, anesthetized mice were subjected to in situ liver perfusion via the portal vein with $Ca^{2+}$-free HBSS for 15 min, followed by sequential perfusion with 0.2% pronase (Aladdin, China, Cat. No. P304928) and 0.2% type IV collagenase (BioFroxx,Germany, Cat. No. 2091MG100) until complete liver blanching and enzymatic digestion were achieved. The cell suspension was filtered through a 100 µm nylon mesh and centrifuged at 50 × g; the cell pellet was processed for hepatocyte isolation, while the supernatant was further subjected to density gradient centrifugation to isolate primary HSCs and macrophages. Both primary cell populations were cultured in DMEM medium containing 10% heat-inactivated FBS. Hepatocytes were verified by their typical polygonal morphology under phase-contrast microscopy. HSCs were identified by their characteristic vitamin A autofluorescence under ultraviolet excitation, a well-established criterion for confirming quiescent HSCs. Macrophages were characterized by immunofluorescence staining using an F4/80-specific antibody.

## RNA extraction and real-time PCR

Total RNA was isolated from samples using TRIzol reagent (TaKaRa, Japan, Cat. No. 9109), according to the manufacturer's instructions. cDNA was synthesized using the HiScript II Q RT Super Mix for qPCR (Vazyme Biotech, China, Cat. No. Q311-03). Gene expression levels were quantified via SYBR Green-based qPCR with the Applied Biosystem 7300 Fast Real-time PCR System, and relative RNA amount was calculated by 2-ΔΔCt method with the normalization to *GAPDH* or *Actb* as internal reference genes. All primers used in this study are listed in S1 Table.

## Western blotting

Total protein was extracted from tissues or cells using pre-cooled RIPA buffer (Beyotime, China, Cat. No. P0038) supplemented with a protease inhibitor cocktail (Thermo Scientific, USA, Cat. No. 87786). Protein concentration was quantified using the Bicinchoninic Acid Protein Assay Kit (Thermo Scientific, USA, Cat. No. A65453). Equal amounts of protein lysate were separated by 10% SDS-PAGE and then transferred to 0.45 µm PVDF membranes (Millipore, USA, Cat. No. IPVH85R). After blocking with 5% non-fat milk in TBST for 1 h at room temperature, the membranes were incubated with corresponding primary antibodies at 4 °C overnight. Following three 10-min washes with TBST, the membranes were incubated with HRP-conjugated secondary antibodies for 1 h at room temperature. The membranes were then washed an additional three times with TBST (10 min each), prior to incubation with ECL Western Blotting Substrate (Bio-Rad, USA,

Cat. No. 1705060) and subsequent exposure to X-Ray Super RX Films (Fujifilm, Japan). All antibodies used in this study are listed in S2 Table.

## Histology and immunohistochemistry

Liver samples were fixed with 4% paraformaldehyde and embedded in paraffin. Serial paraffin sections were stained with Mayer's hematoxylin and eosin (H&E) stain, and hepatic granuloma size was quantified using a calibrated ocular micrometer. Additionally, Masson's trichrome staining was performed on adjacent sections to assess the degree of liver fibrosis. For immunohistochemical (IHC) staining, 4 μm thick liver sections were deparaffinized in xylene (twice, 10 min each) and hydrated through a graded ethanol series followed by deionized water. Microwave-assisted antigen retrieval was performed using EDTA (Beyotime, China, Cat. No. C0167) buffer. Sections were incubated with 3% $H_2O_2$ for 10 min at room temperature, then incubated with the corresponding primary antibodies at 4 °C overnight. Subsequently, sections were incubated with HRP-conjugated secondary antibody for 1 h at room temperature. Antigen-antibody complexes were visualized using 3,3'-diaminobenzidine (DAB, Beyotime, China, Cat. No. P0201M) chromogenic substrate, and cell nuclei were counterstained with hematoxylin. Sections were then dehydrated through a graded ethanol series, cleared in xylene, and mounted in neutral balsam. IHC staining was examined and imaged under a light microscope. All antibodies used for these assays are listed in S2 Table.

## Methylated RNA immunoprecipitation sequencing (MeRIP-seq) and MeRIP-Real-time PCR

Hepatic cells with stable TTP knockdown and control hepatic cells were collected for subsequent assays. MeRIP-sequencing (MeRIP-seq) and downstream data analysis were performed by Seqhealth Technology Co., Ltd. (Wuhan, China). Briefly, 50 μg total RNAs were used for polyadenylated RNA enrichment by VAHTS mRNA Capture Beads (VAHTS, Cat. No. N401-01/02). mRNA was incubated with 20 mM $ZnCl_2$ at 95 °C for 5–10 min to generate RNA fragments predominantly ranging from 100 to 200 nt in length. Following fragmentation, 10% of the RNA fragments were reserved as the "Input" fraction, and the remaining 90% were subjected to $m^6A$ immunoprecipitation (IP). $m^6A$ IP was performed using a specific anti-$m^6A$ antibody (Synaptic Systems, Germany, Cat. No. 202203) provided by Wuhan Seqhealth Co., Ltd. RNA from both Input and IP fractions was extracted using TRIzol reagent (Invitrogen, USA, Cat. No. 15596026). The stranded RNA sequencing library was constructed with KC-Digital Stranded mRNA Library Prep Kit for Illumina (Cat. No. DR08502, Wuhan Seqhealth Co., Ltd. China) in accordance with the manufacturer's protocol. This kit mitigates PCR and sequencing duplication bias by incorporating 8-base random unique molecular identifiers (UMIs) to label pre-amplified cDNA molecules. The library products corresponding to 200–500 bps were enriched, quantified and finally sequenced on an Illumina NovaSeq 6000 platform using the PE150 sequencing mode.

MeRIP-qPCR was performed as previously described [36]. Briefly, total RNA was extracted using TRIzol (TaKaRa, Japan, Cat. No. 9109). Following the reservation of 500 ng total RNA as the Input fraction, the remaining 10 μg RNA was subjected to $m^6A$ immunoprecipitation (IP) with a specific anti-$m^6A$ antibody (Epigentek, USA, Cat. No. A-1801–020) or rabbit IgG (ABclonal, China, Cat. No. AC005, negative control) in IP buffer (150 mM NaCl, 0.1% NP-40, 10 mM Tris at pH 7.4, 100 U RNase inhibitor). $m^6A$-modified RNA was immunoprecipitated using Dynabeads Protein A (Thermo Fisher, USA, Cat. No. 10006D), followed by two rounds of elution with 6.7 mM $N^6$-methyladenosine 5'-monophosphate sodium salt at 4 °C for 1 hour per elution. Target RNA was precipitated overnight at -80 °C by adding 5 μg glycogen, one-tenth volume of 3 M sodium acetate, and 2.5 volumes of 100% ethanol. Finally, $m^6A$ enrichment of target RNAs was quantified by quantitative Real-time PCR, consistent with the real-time PCR protocol described above.

## Co-immunoprecipitation coupled with mass spectrometry (Co-IP-MS) and co-immunoprecipitation (Co-IP)

HEK293T cells transfected with the pHAGE-TTP vector (FLAG-tagged) for TTP expression were cultured for 48 h, For co-immunoprecipitation (Co-IP), $1 \times 10^6$ cells were lysed in 1 mL of lysis buffer (Beyotime, China, Cat. No. P0013J)

supplemented with protease inhibitors on ice for 30 min, followed by centrifugation at 12,000 rpm for 10 min at 4 °C. The supernatant was incubated with appropriate Anti-FLAG Affinity Gel (Biomake, China, Cat. No. B23101) with gentle rotation overnight at 4 °C, After three washes with ice-cold PBST, the beads were boiled with 80 μL of 2 × SDS loading buffer for 5 min. Subsequently, 10 μL of supernatant was collected for sodium dodecyl sulfate-polyacrylamide gel electrophoresis (SDS-PAGE), and the protein bands were visualized by silver staining. The remaining eluate was subjected to mass spectrometry (MS) for subsequent protein analysis.

Cells were lysed in cold IP buffer (150 mM NaCl, 25 mM Tris-HCl, pH 8.0, 1 mM EDTA, 1% Triton X-100, 0.5% Nonidet P-40) containing a protease inhibitor cocktail. The lysates were centrifuged at 12,000 × g for 30 min at 4°C, and the resulting supernatants were pre-cleared with protein A/G agarose beads (Smart-Lifesciences, China, Cat. No. SM005002) at 4°C for 6 h. Subsequently, the pre-cleared supernatants were incubated with FLAG antibody (Proteintech, USA, Cat. No. 66008–4-Ig) or an isotype-matched control IgG overnight at 4°C. Pre-equilibrated protein A/G-agarose beads were then added to precipitate FLAG-tagged protein complexes, followed by incubation at 4°C for 6 h. The immunoprecipitates were washed extensively with cold PBST, eluted by heating in 1 × loading buffer at 95°C for 5 min, and analyzed by immunoblotting. All antibodies used in these assays are listed in S2 Table.

## m⁶A dot blotting

Dot blots were performed using mouse monoclonal anti-m$^6$A antibodies (Epigentek, USA, Cat. No. A-1801–020). A total of 500 ng RNA was denatured at 75 °C for 5 min, immediately placed on ice for rapid cooling, and spotted onto Hybond N+ nylon membranes (GE Amersham, USA, Cat. No. RPN303B). The RNA was then cross-linked to the membranes using a UV Stratalinker 2400 (Analytik Jena US LLC) at 254 nm with an irradiation energy of 150 mJ/cm² for 5 min. Membranes were blocked with 5% nonfat milk in PBS–0.1% Tween 20 at room temperature for 1 h, followed by incubation with the primary anti-m$^6$A antibody at 4 °C overnight. Thereafter, membranes were incubated with HRP-conjugated secondary antibody (Proteintech, USA, Cat. No. HRP-60004) at 4°C overnight. Methylene blue (MB) staining was performed as the loading control for all dot blot assays.

## Luciferase reporter assay

Cells were seeded at a density of $1 \times 10^5$ cells per well in 24-well plates and cultured for 24 h prior to transfection. The pMIR-GLO reporter vectors containing the TGF-β1–5′UTR or MYB-CDS fragment were transfected into the cells using 1 μL Neofect DNA transfection reagent (Neofect, China, Cat. No. TF201201) in accordance with the manufacturer's instructions. After 24 h of transfection, cells were collected into reporter lysis buffer (Promega, USA, Cat. No. E1910), and luciferase activity was detected using a luciferase assay system (Promega, USA). Firefly luciferase activity was normalized to the Renilla luciferase activity for each sample to correct for transfection efficiency variations.

## Chromatin immunoprecipitation assay

ChIP assays were performed as previously described [14]. Briefly, cells were cross-linked with 1%(v/v) paraformaldehyde at room temperature for 10 min, and the cross-linking reaction was quenched by adding glycine (final concentration: 125 mM) followed by incubation for an additional 5 min at room temperature. After cell lysis, the fixed chromatin was fragmented into 100–500 bp DNA fragments using a Diagenode Bioruptor Plus. The DNA fragments were immunoprecipitated overnight at 4 °C with gentle rotation using the indicated antibodies and Protein A/G beads (Smart-Lifesciences, China, Cat. No. SM005002). The anti-SMAD2/3 antibodies used for immunoprecipitation were commercially available (R&D Systems, USA, Cat. No. AF3797). The precipitated DNA samples were analyzed by real-time PCR.

## Adeno-associated viral vectors production

The recombinant AAV serotype vector was a kind gift from Y.C. Xia (Wuhan University). Recombinant AAV vectors expressing the full-length TTP CDS (AAV-TTP) or a TTP-targeting shRNA (AAV-shTTP) were constructed and verified by sequencing. These recombinant plasmids were subsequently packaged into infectious AAV particles via a helper-free triple-transfection method as previously described, and the viral particles were purified according to a published protocol [14]. Briefly, HEK293T cells were co-transfected with AAV-DJ (Addgene, Cat. No. 130878), px552 (Addgene, Cat. No. 60958), and AAV-helper (Addgene, Cat. No. 81070) plasmids in 150-mm dishes at 80% confluence. Viral titers were quantified by real-time PCR. The purified recombinant AAV virions were resuspended in PBS at a final titer of $1 \times 10^{11}$ viral particles per milliliter (vp/mL).

## Statistics

All experimental data are presented as the mean±standard deviation (SD), derived from at least three independent biological replicates. Investigators were blinded to group assignments throughout the process of outcome assessment. Sample sizes were calculated a priori to ensure adequate statistical power for identifying the prespecified effect size. Statistical analyses were performed with two-tailed Student's t-test, one-way ANOVA, or two-way ANOVA, as appropriate for the experimental design. Spearman's rank correlation analysis was utilized to compute correlation coefficients (r). A value of $P < 0.05$ was regarded as statistically significant.

## Supporting information

**S1 Fig. The expression level of TTP correlated with that of fibrotic factors.** (A-C) Spearman correlation analysis was applied to assess the correlations between *TTP* expression and that of *COL1A1* (A), *COL4A1* (B), and *TGFB1* (C) in human liver samples with *S. japonicum*-induced liver fibrosis, using data from the GSE61376 dataset. (D) The mRNA expression of TTP in hepatocytes (HC), hepatic stellate cells (HSC), and macrophages isolated from mouse liver samples. (E) Mouse primary hepatocytes were exposed to SEA (60 μg/ml) for 24 h followed by real-time PCR analysis for *Ttp* mRNA. Data represent mean±SD from three independent experiments. Statistical analyses were performed using one-way ANOVA or 2-tailed Student's t-test. *$P < 0.05$ versus ctrl.
(TIF)

**S2 Fig. The expression of *Ttp, Acta2, Col1α1,* and *Col3α1* in livers from non-infected and *S. japonicum*-infected mice after PBS, AAVDJ-Ctrl, or AAVDJ-shTTP injection.** (A) Western blot (left) and real-time PCR (right) assays were conducted to assess TTP expression in liver tissues of non-infected mice following 2 weeks of AAV vector-mediated (AAVDJ-shTTP) TTP knockdown. (B) H&E staining and Masson's trichrome staining of liver sections from the indicated groups (PBS/infected and AAVDJ-shCtrl/infected) (Scale bar: 200 μm). (C) Granuloma size was measured from H&E-stained liver sections, and fibrosis score was determined from Masson's trichrome-stained liver sections. (D) The content of hydroxyproline in the liver was detected. (E) Real-time PCR analysis of *Acta2, Col1α1,* and *Col3α1* expression level in liver tissues. Data represent mean±SD from three independent experiments. Statistical analyses were performed using 2-tailed Student's t-test. *$P < 0.05$ versus ctrl. ns, not significant.
(TIF)

**S3 Fig. Cytokine mRNA expression in livers from *S. japonicum*-infected mice after AAVDJ-Ctrl, or AAVDJ-shTTP injection.** (A) Real-time PCR analysis of *Il6, Il8,* and *Ccl2* expression level in liver tissues. (B) Real-time PCR analysis of *Ifng* and *Tnf* expression level in liver tissues. (C) Real-time PCR analysis of *Il4* and *Il13* expression level in liver tissues. (D) Real-time PCR analysis of *Il17a* expression level in liver tissues. Data represent mean±SD from three independent experiments. Statistical analyses were performed using 2-tailed Student's t-test. *$P < 0.05$ versus ctrl.
(TIF)

**S4 Fig. The expression of *Ttp*, *Acta2*, *Col1α1*, and *Col3α1* in livers from non-infected and *S. japonicum*-infected mice after PBS, AAV8-Ctrl, or AAV8-TTP injection.** (A) Western blot (left) and real-time PCR (right) assays were conducted to assess TTP expression in liver tissues of non-infected mice following 2 weeks of AAV8 vector-mediated (AAV8-TTP) TTP overexpression. (B) H&E staining and Masson's trichrome staining of liver sections from the indicated groups (PBS/infected and AAV8-Ctrl/infected) (Scale bar: 200 μm). (C) Granuloma size was measured from H&E-stained liver sections, and fibrosis score was determined from Masson's trichrome-stained liver sections. (D) The content of hydroxyproline in the liver was detected. (E) Real-time PCR analysis of *Acta2*, *Col1α1*, and *Col3α1* expression level in liver tissues. Data represent mean±SD from three independent experiments. Statistical analyses were performed using 2-tailed Student's t-test. *$P<0.05$ versus ctrl. ns, not significant.
(TIF)

**S5 Fig. TTP negatively modulates the expression of fibrosis-related cytokines.** (A) The heat map illustrates the gene expression profiles of fibrosis-related cytokines in TTP-knockdown cells based on our previous RNA-seq data (GSE157581). (B) The mRNA levels of *TGFB1*, *TIMP1*, *COL1A*, *COL4A1*, and *MYB* were analyzed in TTP-knockdown hepatic cells or LX-2 cells via real-time PCR. (C) The mRNA levels of *TGFB1*, *TIMP1*, *COL1A*, *COL4A1*, and *MYB* was analyzed in TTP-overexpression hepatic cells or LX-2 cells via real-time PCR. Data represent mean±SD from three independent experiments. Statistical analyses were performed using 2-tailed Student's t-test and one-way ANOVA. *$P<0.05$ versus ctrl.
(TIF)

**S6 Fig. m⁶A RNA modification modulates the stabilization of TGF-β1 and MYB mRNAs.** (A) An integrative genomics viewer (IGV) tracks display higher enrichment of m⁶A peaks in *TGFB1* and *MYB* transcripts. (B) The mRNA levels of *TGFB1* and *MYB* in HepG2 cells treated with 3-DAA were measured by real-time PCR. (C) The stability of *TGFB1* and *MYB* mRNAs in hepatic cells treated with 3-DAA. (D) Analysis of luciferase levels of pMIR-TGF-β1–5'UTR, pMIR-MYB-CDS, pMIR-TGF-β1–5'UTR mut, or pMIR-MYB-CDS mut in hepatic cells treated with 3-DAA. Data represent mean±SD from three independent experiments. Statistical analyses were performed using 2-tailed Student's t-test and two-way ANOVA. *$P<0.05$ versus ctrl.
(TIF)

**S7 Fig. TTP negatively regulates TGF-β1 and MYB expression by activating WTAP.** (A) Hepatic cells overexpressing METTL14 or YTHDF2 were transfected with pMIR-TGF-β1–5'UTR or pMIR-MYB-CDS, followed by luciferase analysis. (B) Stable WTAP-knockdown LX-2 cells were transfected with pMIR-MYB-CDS or pMIR-MYB-CDS mut for 24 h, followed by luciferase analysis. (C)TTP and SMAD2/3 were immunoprecipitated in HEK293T cells. (D) HEK293T cells overexpressing SMAD2/3 were transfected with pGL-YTHDF2, followed by luciferase analysis. (E) HEK293T cells were co-transfected with the promoters of WTAP or METTL14, along with TTP-N, TTP-Zn, or TTP-C, followed by luciferase analysis. (F) The expression of TTP, WTAP, and METTL14 in hepatic cells with TGF-β stimulation. Data represent mean±SD from three independent experiments. Statistical analyses were performed using 2-tailed Student's t-test and one-way ANOVA. *$P<0.05$ versus ctrl. ns, not significant.
(TIF)

**S8 Fig. TTP represses HSC activation via regulating TGF-β1 expression.** (A) The expression of *SMAD7*, *COL1A1*, and *ACTA2* in LX-2 cells was analyzed by real-time PCR after 36 h of coculture with hepatic cells stably expressing shTTP, in the presence or absence of a neutralizing antibody to TGF-β1. (B) The expression of *SMAD7*, *COL1A1*, and *ACTA2* was analyzed in LX-2 cells after coculture for 36 h with stably WTAP-knockdown hepatic cells transfected with TTP-overexpressing vectors. Data represent mean±SD from three independent experiments. Statistical analyses were performed using 2-tailed Student's t-test. *$P<0.05$ versus ctrl.
(TIF)

**S9 Fig. The level of m⁶A RNA modification in liver fibrosis induced by *S.japonicum* infection.** (A) Dot blot was used to measure m⁶A RNA modification in the livers of *S. japonicum*-infected mice. (B) Immunohistochemical (IHC) analysis was performed to evaluate WTAP, METTL14, and YTHDF2 expression in fibrotic liver tissues from mice infected with *S. japonicum*. (Scale bar: 200 μm). (C) Mouse primary hepatocytes and LX-2 cells were treated with SEA at concentrations of 60 μg/mL and 30 μg/mL, respectively, for 24 h, followed by real-time PCR analysis for *TGFB1*, *MYB*, *WTAP*, *METTL14*, and *YTHDF2* mRNA. (D) H&E staining of liver sections from mice induced by 3-DAA (1.5 μg/g) (scale bar: 100 μm). (E) Plasma ALT and AST levels in mice. Data represent mean ± SD from three independent experiments. Statistical analyses were performed using 2-tailed Student's t-test. *$P < 0.05$ versus ctrl. ns, not significant. (TIF)

**S1 Table. Primers and shRNA sequences.** (DOCX)

**S2 Table. The information of primary antibodies.** (DOCX)

**S3 Table. Proteins interacting with TTP in HEK293T cells as determined by mass spectrometry analysis of FLAG or IgG immunoprecipitations.** (XLSX)

## Acknowledgments

Recombinant AAV vectors were kindly provided by Prof. Y.C. Xia (Wuhan University).

## Author contributions

**Conceptualization:** Xuejun Zhao, Xiaoqing Li, Huifen Dong, Rui Zhou.

**Data curation:** Xuejun Zhao, Xinyi Wang, Kejun Liu, Xiaoqing Li, Huifen Dong, Rui Zhou.

**Formal analysis:** Xuejun Zhao, Xinyi Wang, Kejun Liu, Jiaxi Zhang, Jiayi Feng, Qiong Luo, Lanxuan Jiang, Rui Zhou.

**Funding acquisition:** Xuejun Zhao, Qiong Luo, Huifen Dong, Rui Zhou.

**Investigation:** Xuejun Zhao, Xinyi Wang, Kejun Liu, Qian Fang, Cheng Wang, Guangbo Mei, Lihua Qu, Jiayi Feng, Xiaoqing Li, Huifen Dong, Rui Zhou.

**Methodology:** Xuejun Zhao, Xinyi Wang, Kejun Liu, Qian Fang, Cheng Wang, Guangbo Mei, Lihua Qu, Jiaxi Zhang, Jiayi Feng, Wenjuan Yang, Junnan Zhu, Hang Zhuo, Na Kuang, Xuebing Qiu, Zhigao Deng, Qiong Luo, Xiaoxia Jin, Lanxuan Jiang, Xiaoqing Li, Huifen Dong, Rui Zhou.

**Project administration:** Rui Zhou.

**Resources:** Na Kuang, Rui Zhou.

**Supervision:** Xiaoqing Li, Huifen Dong.

**Validation:** Xinyi Wang, Qian Fang, Jiayi Feng, Wenjuan Yang, Junnan Zhu, Xuebing Qiu, Xiaoxia Jin, Xiaoqing Li, Huifen Dong, Rui Zhou.

**Visualization:** Rui Zhou.

**Writing – original draft:** Xuejun Zhao, Xiaoqing Li, Huifen Dong, Rui Zhou.

**Writing – review & editing:** Xuejun Zhao, Xinyi Wang, Qian Fang, Lanxuan Jiang, Xiaoqing Li, Huifen Dong, Rui Zhou.

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
