## [Decision Letter · Decision Letter 0]

12 Sep 2025

PPATHOGENS-D-25-01986

TTP alleviates schistosomiasis-induced liver fibrosis by regulating TGF-β1 and MYB stability in an m6A-dependent manner

PLOS Pathogens

Dear Dr. Zhou,

Thank you for submitting your manuscript to PLOS Pathogens. After careful consideration, we feel that it has merit but does not fully meet PLOS Pathogens’s publication criteria as it currently stands. Therefore, we invite you to submit a revised version of the manuscript that addresses the points raised during the review process.

Please submit your revised manuscript within 60 days by 11/11/25. If you will need more time than this to complete your revisions, please reply to this message or contact the journal office at plospathogens@plos.org. When you’re ready to submit your revision, log on to https://www.editorialmanager.com/ppathogens/ and select the ’Submissions Needing Revision’ folder to locate your manuscript file.

* A rebuttal letter that responds to each point raised by the editor and reviewer(s). You should upload this letter as a separate file labeled ’Response to Reviewers’. This file does not need to include responses to any formatting updates and technical items listed in the ’Journal Requirements’ section below.

* A marked-up copy of your manuscript that highlights changes made to the original version. You should upload this as a separate file labeled ’Revised Manuscript with Track Changes’.

* An unmarked version of your revised paper without tracked changes. You should upload this as a separate file labeled ’Manuscript’.

We look forward to receiving your revised manuscript.

Kind regards,

David L. Williams

Academic Editor

PLOS Pathogens

Edward Mitre

Section Editor

PLOS Pathogens

Sumita Bhaduri-McIntosh

Editor-in-Chief

PLOS Pathogens

orcid.org/0000-0003-2946-9497

Michael Malim

Editor-in-Chief

PLOS Pathogens

orcid.org/0000-0002-7699-2064

**Additional Editor Comments:**

The study was appreciated for its novelty and importance by all three reviewers. However, they also identified a number of shortcomings, specifically in the use of appropriate controls, some studies need justification for relevance to schistosome infections, and inadequate image analyses. Experimentation is needed to address these issues. Please address specifically, but not limited to:

Experimental approach needs to be more tightly linked to liver pathology due to schistosome infection and not general liver damage as indicated by reviewer 2 in multiple comments.

Reviewer 3, comment 5 requests that a number of statistical determinations need to be clearly stated. It appears that n=6 was recorded for most measurements; given the number of granulomas in a liver, n=6 is not adequate.

More details on the dosing regimen and pharmacological properties of DAA should be provided.

The nature of the analyses of the images in fig 1C needs to be addressed.

The controls for figures 2 and 3 are not adequate as indicated by reviewer 2. This needs to be addressed.

The authors are strongly required to thoroughly revise the manuscript for clarity, ideally with assistance from a native English speaker or professional language editing service.

**Journal Requirements:**

At this stage, the following Authors/Authors require contributions: Cheng Wang. Please ensure that the full contributions of each author are acknowledged in the "Add/Edit/Remove Authors" section of our submission form.

- ® on page: 26.

3) Please amend your detailed Financial Disclosure statement. This is published with the article. It must therefore be completed in full sentences and contain the exact wording you wish to be published.

1) State what role the funders took in the study. If the funders had no role in your study, please state: "The funders had no role in study design, data collection and analysis, decision to publish, or preparation of the manuscript.".

**Reviewers’ Comments:**

Reviewer’s Responses to Questions

**Part I - Summary**

Reviewer #1: In this study, the authors investigated the potential role of Tristetraprolin (TTP) in the pathogenesis of Schistosoma japonicum-induced liver fibrosis. Using a murine model, they first demonstrated that TTP expression is upregulated at both the RNA and protein levels in the liver following infection. Notably, overexpression of TTP ameliorated infection-associated liver fibrosis. To further elucidate the underlying mechanisms, the authors employed various cell lines and found that TTP negatively regulates the mRNA stability of TGF-β1 and MYB by promoting N6-methyladenosine (m6A) RNA methylation, thereby inhibiting hepatic stellate cell activation. Additionally, TTP was shown to activate transcription of WT1-associated protein (WTAP) through its interaction with SMAD2/3. Furthermore, using an m6A RNA methylation inhibitor, the study demonstrated that TTP protects against S. japonicum-induced liver fibrosis in vivo by enhancing m6A RNA methylation. These findings reveal novel mechanisms involving TTP-mediated m6A RNA hypermodification in the pathogenesis of S. japonicum-induced liver fibrosis, with potential implications for the development of future therapeutic strategies. Overall, the experiments were well-performed, making the study of interest to the journal’s readership. However, the manuscript suffers from a general lack of clarity, as detailed below, which needs to be addressed.

Reviewer #2: -Well written and easy to read.

-The quality of all figures was very low (blurry). Fonts small and fuzzy/blurry. Hard to read/see in many instances and appreciate the data. The blurry/ fuzziness was not an issue in the supplemental file, so that might be an issue with the Pdf file for the main text and should be corrected.

-In introduction, there is no information and background on the importance of TGF-β1 and MYB in schistosome/schistosome induced fibrosis and whether there is any previous literature. This is important since the manuscript is focused on the regulation of these molecules.

-Figure legends: No information regarding how many times experiments have been performed, biological repeats and technical repeats. This is the problem for all figures in the main text and supplementary figures.

-One concerning issue is that, while many pathways discussed could be relevant and compelling in the broader context of liver fibrosis, their applicability weakens when the authors attempt to connect them specifically to schistosomiasis-related liver fibrosis (see details in Part II).

-Authors do not discuss the cellular source of TTP anywhere in the manuscript and what cells they think might be the source in the context of schistosome in the liver.

Reviewer #3: This study presents a comprehensive investigation into the role of Tristetraprolin (TTP) in Schistosoma japonicum-induced liver fibrosis, with a focus on its regulation of m6A RNA methylation through interactions with SMAD2/3 and subsequent effects on TGF-β1 and MYB expression. The topic is novel and highly relevant to the field of hepatology and parasitology. The experimental design is generally rigorous, incorporating in vivo models, multi-omics approaches, and detailed mechanistic insights. However, several critical issues pertaining to methodological justification, analytical quantification, and experimental controls need to be addressed to fully strengthen the validity and impact of the conclusions.

**Part II – Major Issues: Key Experiments Required for Acceptance**

Reviewer #1: Important modifications are required, as detailed below in Part III.

Reviewer #2: -For in vivo experiments in figures 2 and 3 weights of infected livers should be included. Also weights of uninfected livers as control should be included.

-Fig 2D and Fig. 3D and Fig. 8 show the granuloma size which is significant, but it is not enough. Authors need to show the total numbers of eggs in liver tissue (per gr or per field) as well to make sure that the parasite load was the same.

-Line 178: authors state “To systematically elucidate the role of TTP in liver fibrosis, we reanalyzed our previously published RNA-Seq dataset from TTP-knockdown cells (GSE157581)”. The study that they are referring to is acute liver failure and is different from liver fibrosis associated with schistosomiasis. They found these 2 molecules TGF-β1 and MYB from this RNA-Seq dataset and they validated using qPCR in hepatocytes and LX-2 cells but none were performed in cells stimulated with soluble egg antigens (SEA) or schistsome eggs, making it relevant to liver fibrosis associated with schistosome infection.

-Similar to the comment above, experiments in Figures 6 and 7 are interesting per say but none are performed in the context of schisto, which could have been done in the presence of eggs or SEA. For example, the authors show that TGF-β treatment significantly increased the mRNA expression of WTAP and METTL14 at the mRNA level (Fig. S3G), but what does that mean in the context of schisto infection and/or upon egg stimulation in vitro which is the focus of this manuscript?

-Line 65: “our study has revealed TTP alleviates schistosomiasis-induced liver fibrosis by regulating TGF-β1 and MYB mRNA stability in an m6A-dependent manner.” For reasons mentioned above this sentence needs to be corrected

-Fig. 8 shows that m6A RNA methylation contributes to TTP-mediated protection against liver fibrosis but it does not show that it is via regulating TGF-β1 and MYB stability, and the title of the manuscript “TTP alleviates schistosomiasis-induced liver fibrosis by regulating TGF-β1 and MYB stability in an m6A-dependent manner” needs to be corrected.

- Confocal image in Fig. 1C is not convincing. Fig. 1C: middle a-SMA (there is a lot of green that is not merged with red). I see more colocalization with AJB (top) than a-SMA-Bottom. Qualification could be very helpful.

There is also staining for F480 but there is no explanation in the text regarding F480 staining and why they included it.

- Authors show that TTP modulates HSC activation by regulating TGF-β1 expression (Fig. S4A). How about a role for MYB? Since it is also included in the title of the manuscript? Did the authors investigate the role for this molecule? Any previous literature regarding this? It is important that authors discuss this.

Reviewer #3: 1. Quantification of Immunofluorescence (IF) Data: While Figure 1C suggests TTP is "predominantly localized in hepatocytes," the claim requires robust quantitative support. The authors should perform and report quantitative image analysis (e.g., Pearson’s correlation coefficient for co-localization with cell-specific markers, or mean fluorescence intensity measurements across different cell types) to objectively substantiate this key observation. Additionally, quantifying TTP mRNA or protein levels in sorted populations of hepatocytes, hepatic stellate cells (HSCs), Kupffer cells, and endothelial cells from fibrotic livers would provide definitive evidence and greatly strengthen the foundation of the study.

2. Rationale for Focus on Hepatocytes: The study focuses on hepatocytes as the primary functional site of TTP, despite also detecting TTP in HSCs (Fig. 1C). The rationale for prioritizing hepatocytes over HSCs is not adequately explained. Given that HSCs are the central effector cells in fibrosis, the authors should either:

Provide functional data comparing TTP effects in hepatocytes vs. HSCs, or

Clearly justify their focus on hepatocytes and discuss the potential paracrine mechanisms by which hepatocyte TTP influences HSC activation.

3. Justification for Experimental Timeline: The choice to administer AAV vectors at day 10 post-infection is a critical methodological detail that lacks justification. Please provide a scientific rationale for this specific timepoint.

4. In Vivo Control Group: The authors have convincingly demonstrated that TTP overexpression via AAV8 delivery ameliorates liver fibrosis. However, the design of the in vivo experiment lacks a necessary control group to fully validate the findings. The current study includes only two groups: AAV8-Ctrl and AAV8-TTP. The AAV8-Ctrl group is intended to serve as the control, but it is not sufficient to rule out the potential effects of the AAV8 viral vector itself on the progression of schistosomiasis-induced liver fibrosis. It is well-known that viral vectors can sometimes elicit innate immune responses or other off-target effects that might inadvertently influence the disease model. Therefore, to unequivocally attribute the observed therapeutic effects solely to TTP overexpression, a third control group—a non-treatment group (e.g., injected with PBS or saline)—is essential. This would allow the authors to confirm that any difference between the AAV8-TTP and AAV8-Ctrl groups is specifically due to the action of TTP.

5. Statistical Transparency: For all histological data (e.g., granuloma area, fibrosis area in Figs. 2C-D, 3C-D, 8D-E), the number of fields quantified, biological replicates (n), and statistical methods should be clearly stated in the figure legends.

6. Clinical Correlation Analysis: The use of human transcriptomic data (GSE61376) to show TTP upregulation is a strength. To further enhance the clinical relevance, the authors could explore whether TTP expression levels correlate with established markers of fibrosis severity (e.g., collagen gene expression, AST/ALT levels if available) within this dataset.

7. Schematic Diagram: A graphical abstract or a summary schematic model illustrating the proposed TTP-SMAD2/3-WTAP-m6A-TGF-β1/MYB-HSC activation axis would greatly aid readers in visualizing and understanding the complex mechanistic pathway proposed by this study.

**Part III – Minor Issues: Editorial and Data Presentation Modifications**

Reviewer #1: 1. Many sentences and statements throughout the manuscript are confusing or ambiguous and require revision for clarity. For example:

• Lines 41–42: “TTP is upregulated in S. japonicum-induced liver fibrosis” — it is unclear in which tissue this upregulation occurs. Please specify.

• Lines 46–47: “TTP activates WATP transcript” — do the authors mean that TTP activates WTAP mRNA molecule or transcription of the WTAP gene? This should be clearly stated.

• Lines 59–60: “the mechanism underlying the fine-tuning regulation of inflammatory cytokines and chemokines” — it is unclear whether this refers to regulation of their gene expression or their biological function.

• The authors are strongly encouraged to thoroughly revise the manuscript for clarity, ideally with assistance from a native English speaker or professional language editing service.

2. The description of the results in Figure 1A should clearly indicate that TTP expression is being assessed at the protein level. Additionally, in lines 129–133, the current wording implies that TTP, rather than albumin, is predominantly located in hepatocytes. This should be corrected to accurately reflect that albumin is the hepatocyte marker, and TTP’s localization is being evaluated in relation to it in hepatocytes, as well as in HSCs.

3. The result descriptions for Figures 2 and 3 should include comparisons between TTP knockdown or overexpression and the corresponding controls from non-infected animals. Currently, the descriptions focus primarily on infected animals, which limits the interpretation of TTP’s specific role in infection-induced changes. Including these comparisons would provide a more complete understanding of TTP’s effects under both baseline and infected conditions.

4. In Figure 6A, it would be important to clarify whether TTP can directly interact with WTAP. Was WTAP identified in the mass spectrometry (MS) data? If so, this should be explicitly stated. If WTAP was not detected in the MS results, this negative finding is also worth mentioning, as it may suggest that the interaction is indirect or occurs under specific conditions not captured in the assay.

5. Several other proteins identified in the list may also function as transcriptional activators. Therefore, the authors should provide a clear rationale for selecting SMAD2/3 for further analysis. Explaining this choice—whether based on prior literature, expression patterns, relevance to liver fibrosis, or preliminary data—would strengthen the interpretation of the results and the logic behind the experimental design.

6. It is unclear whether the DAA inhibitor was administered as a single intraperitoneal injection at 42 days post-infection. If so, the authors should clarify the duration of DAA’s activity in vivo, as it is questionable whether a single dose would remain effective for an extended period. If possible, additional details regarding the dosing regimen, half-life, and pharmacokinetics of DAA in this model would be helpful for evaluating the validity of the experimental approach.

7. In lines 356–357, the text incorrectly refers to SMAD2/3 as an important m6A writer. This should be corrected, as it is WTAP, not SMAD2/3, that functions as a core component of the m6A methyltransferase complex. Please revise this statement to accurately reflect the roles of these proteins.

8. Including a schematic diagram or cartoon summarizing the proposed pathway would greatly enhance the manuscript’s clarity and impact. For example, a visual representation illustrating the sequence: infection → TTP induction → interaction with SMAD2/3 to increase WTAP transcription → enhanced m6A methylation → regulation of TGF-β and MYB mRNA stability → hepatic stellate cell (HSC) activation → liver fibrosis would help readers better understand the complex molecular mechanisms described.

Reviewer #2: -Minor comment -Line 87: Authors mention that TTP negatively regulates several proinflammatory cytokines such as IL-6, IL-8 and CCL2. Since these proinflammatory molecules have an important role in schistosomiasis, it makes sense that authors would investigate whether TTP also affects these cytokines and chemokines during schistosome infection.

-This is a minor comment and would make the manuscript stronger if included: How about the cytokine profile of the liver of these mice? Would be a great addition to the manuscript to assess a few Th1 and Th2 and Th17 cytokines.

-Figure 5: Authors already have reported that TTP modulates m6A modification by regulating the expression of WTAP and in this study (Fig 5) they just investigate whether WTAP was involved in the regulation of TGF-β1 and MYB. I don’t find this novel and can be moved to supplemental. This is a minor comment.

-Please explain in the text that a-SMA is marker for activated HSC

-Line 137: Authors explain data in the text about Fig. 1D which does not exist-Please correct

-Line 203: “Wild” and not “Wile”. Please correct

-Line 339: why the injection of DAA, was performed at 42 days after S. japonicum infection? Would be great if authors describe the rationale or refer to previous literature if any.

-A problem of this work is that no dose response for the inhibitor 3-DAA is used and we don’t know if it is causing cell death or toxicity.

Reviewer #3: Language and Grammar: The manuscript requires thorough proofreading to address numerous grammatical errors and improve clarity. While the scientific content is strong, the presentation occasionally hinders comprehension. I strongly recommend revision by a native English speaker or a professional editing service.

Specific examples include:

Line 111-112: The sentence structure is incorrect and the word order is confusing.

Original: "TTP had been identified to negatively regulate TGF-β1 and MYB by activating WTAP transcription, thereby inhibiting HSCs activation S. japonicum-induced in liver fibrosis."

Suggested Revision: "TTP was identified to negatively regulate TGF-β1 and MYB by activating WTAP transcription, thereby inhibiting HSCs activation in S. japonicum-induced liver fibrosis."

Line 137-139: The word "predominated" is used incorrectly; it should be the adverb "predominantly".

Original: "...indicate that TTP is substantially upregulated predominated in hepatocytes during liver fibrosis."

Suggested Revision: "...indicate that TTP is substantially upregulated, predominantly in hepatocytes, during liver fibrosis."

Line 572: There is a likely spelling error in "stereotype".

Original: "Recombinant AAV stereotype vector was a gift..."

Suggested Revision: "Recombinant AAV vectors of serotype 8 (AAV8) were kindly provided by Prof. Y.C. Xia (Wuhan University)."

PLOS authors have the option to publish the peer review history of their article (what does this mean?). If published, this will include your full peer review and any attached files.

Reviewer #1: No

Reviewer #2: No

Reviewer #3: No

**Figure resubmission:**
---

## [Decision Letter · Decision Letter 1]

4 Feb 2026

PPATHOGENS-D-25-01986R1

TTP alleviates schistosomiasis-induced liver fibrosis via m6A-mediated of regulating TGF-β1 stability mRNA stability

PLOS Pathogens

Dear Dr. Zhou,

Thank you for submitting your manuscript to PLOS Pathogens. After careful consideration, we feel that it has merit but does not fully meet PLOS Pathogens’s publication criteria as it currently stands. Therefore, we invite you to submit a revised version of the manuscript that addresses the points raised during the review process.

* A letter that responds to each point raised by the editor and reviewer(s). You should upload this letter as a separate file labeled ’Response to Reviewers’. This file does not need to include responses to any formatting updates and technical items listed in the ’Journal Requirements’ section below.

* A marked-up copy of your manuscript that highlights changes made to the original version. You should upload this as a separate file labeled ’Revised Manuscript with Track Changes’.

* An unmarked version of your revised paper without tracked changes. You should upload this as a separate file labeled ’Manuscript’.

We look forward to receiving your revised manuscript.

Kind regards,

David L. Williams

Academic Editor

PLOS Pathogens

Edward Mitre

Section Editor

PLOS Pathogens

Sumita Bhaduri-McIntosh

Editor-in-Chief

PLOS Pathogens

orcid.org/0000-0003-2946-9497

Michael Malim

Editor-in-Chief

PLOS Pathogens

orcid.org/0000-0002-7699-2064

**Additional Editor Comments:**

Dr. Zhou, The reviewers have completed their assessment of your manuscript and all are in agreement that the revised manuscript is substantially improved and, with minor editorial changes, it will be acceptable for publication in PLoS Pathogens. I agree with their conclusions. Both R2 and R3 have concerns about the title of the paper, which needs to be changed. R3 has specific comments that should be addressed to clarify the text.

**Journal Requirements:**

1) In the online submission form, you indicated that your data will be submitted to a repository upon acceptance. We strongly recommend all authors deposit their data before acceptance, as the process can be lengthy and hold up publication timelines. Please note that, though access restrictions are acceptable now, your entire minimal dataset will need to be made freely accessible if your manuscript is accepted for publication. This policy applies to all data except where public deposition would breach compliance with the protocol approved by your research ethics board. If you are unable to adhere to our open data policy, please kindly revise your statement to explain your reasoning and we will seek the editor’s input on an exemption.

2) Please amend your detailed Financial Disclosure statement. This is published with the article. It must therefore be completed in full sentences and contain the exact wording you wish to be published.

2) If any authors received a salary from any of your funders, please state which authors and which funders..

**Reviewers’ Comments:**

Reviewer’s Responses to Questions

**Part I - Summary**

Reviewer #1: The authors have adequately addressed all concerns raised in my review.

Reviewer #2: The title of the manuscript has a problem and needs to get fixed. More detail in Part II

Reviewer #3: This manuscript addresses an important and understudied aspect of host–parasite interaction by implicating tristetraprolin (TTP) and m6A RNA methylation in the regulation of Schistosoma japonicum induced liver fibrosis. The topic is relevant to the journal, and the proposed TTP–WTAP–m6A–TGF-β1 axis is conceptually interesting.

However, several mechanistic and conceptual issues must be addressed before this work can be considered for publication in PLOS Pathogens. In its current form, the manuscript overstates causality, lacks sufficient clarity on epitranscriptomic mechanisms, and does not yet fully demonstrate how parasite-driven pathology specifically engages the proposed pathway.

Major Strengths:

Strong Mechanistic Depth: The multi-layered approach, integrating human data, gain/loss-of-function in vivo models, primary cells, and detailed molecular assays, provides compelling support for the proposed pathway.

Technical Rigor: The use of complementary methods to probe m6A (MeRIP-seq, qPCR, dot blot, pharmacological inhibition) strengthens the conclusions.

Clear Disease Relevance: The study convincingly links the molecular findings to HSC activation and fibrosis progression in a relevant infection model.

Well-Integrated Discussion: The discussion effectively contextualizes the findings within the broader field, including a balanced analysis of TTP’s context-dependent roles.

Areas for Improvement (Clarity and Precision):

The primary issues relate to the precision of language and the completeness of mechanistic narrative, not to the validity of the core findings. Specifically, the manuscript would benefit from: 1) clearer phrasing regarding causal relationships, 2) a more detailed conceptual model for m6A-mediated regulation, and 3) tightened narrative focus. These are matters of presentation and interpretation that can be resolved through textual revisions.

Novelty and Significance:

The study makes a valuable contribution by linking TTP’s anti-fibrotic function to the epitranscriptomic machinery (WTAP/m6A) in the context of a parasitic disease. This novel integration of transcriptional co-regulation and post-transcriptional mRNA modification in controlling TGF-β1 output offers fresh insights into the pathogenesis of schistosomiasis liver fibrosis and identifies potential nodes for therapeutic intervention.

**Part II – Major Issues: Key Experiments Required for Acceptance**

Reviewer #1: (No Response)

Reviewer #2: The title has a problem and needs to get fixed: It is currently “TTP alleviates schistosomiasis-induced liver fibrosis via m6A-mediated of regulating TGF-β1 stability mRNA stability” – “stability” is repeated twice and doesn’t make sense in general.

In the rebuttal, the authors mention that they changed the title to this title: “TTP alleviates schistosome-induced liver fibrosis via m⁶A-mediated regulation of TGF-β1 mRNA stability”, but this is different from the title that they used in the manuscript.

The authors have addressed all the previous comments and have added a substantial amount of new data to the manuscript.

Reviewer #3: The following points require clarification and more precise presentation to fully validate the study’s conclusions. These can be addressed through textual modifications and enhanced discussion.

Title and Causality Language: The current title is grammatically problematic. More importantly, throughout the text, some phrasing could be misinterpreted as overstating direct causality (e.g., "TTP protects by increasing m6A"). The data robustly show that m6A is required for TTP’s effect. This important distinction should be reflected in a revised title and precise language.

Suggested Action: Adopt a clearer title (e.g., "Tristetraprolin attenuates schistosomiasis-induced liver fibrosis through m6A-mediated regulation of TGF-β1 mRNA stability"). Systematically review the manuscript to ensure causal language accurately reflects the nature of the data, preferring phrases like "m6A-dependent protection" or "associated with."

Mechanism of WTAP Regulation: The statement that "TTP activates the WTAP gene" is a strong claim. While the ChIP-PCR data are supportive, they do not fully distinguish between direct transcriptional activation and indirect effects.

Suggested Action: Modify the language to more precisely describe the observation (e.g., "TTP promotes WTAP expression" or "is associated with increased WTAP transcription"). The Discussion should acknowledge that the exact mechanism (direct vs. indirect) warrants further investigation, which would strengthen the manuscript’s scholarly tone.

Conceptual Model for m6A Function: The manuscript establishes the necessity of m6A but does not elaborate on the logical downstream mechanism. The fate of m6A-modified mRNA depends on "reader" proteins.

Suggested Action: Expand the Discussion to propose a plausible model. Given the observed mRNA destabilization, the authors should discuss the potential involvement of decay-promoting readers (e.g., YTHDF2) and hypothesize how the TTP/SMAD2/3 complex might interface with this step. This conceptual framing will complete the mechanistic story and demonstrate a deeper level of analysis.

**Part III – Minor Issues: Editorial and Data Presentation Modifications**

Reviewer #1: (No Response)

Reviewer #2: The quality of all figures in the main is still very low. All the main figures are blurry/fuzzy. The supplemental figures are sharp and high quality. This is a minor comment.

Reviewer #3: Addressing these points will enhance the manuscript’s clarity, reproducibility, and overall polish.

Abstract & Introduction:

Soften broad therapeutic claims (e.g., "suggests a potential strategy" vs. "provides a rationale for targeting").

Consider de-emphasizing the mention of MYB in the introduction if it is not central to the main narrative, to maintain focus on the TGF-β1 axis.

Clarify in which specific model prior TTP/m6A findings were made to highlight the novelty of the current work.

Proofread for minor grammatical issues and tighten the final introductory paragraph.

Materials and Methods:

Specify the administration timeline for the inhibitor 3-DAA relative to infection.

Briefly note the assessed purity of isolated primary hepatocytes and HSCs.

Explicitly state the number of biological replicates for MeRIP-seq.

Add a short statement validating the specificity of key antibodies (custom anti-SMAD2/3, commercial m6A).

Results:

Adjust language regarding TTP expression to avoid implying exclusivity to hepatocytes (e.g., "predominantly expressed in...").

Frame MYB clearly as a secondary target rather than a primary driver equivalent to TGF-β1.

When describing the SMAD2/3 interaction, use phrasing like "TTP acts as a transcriptional co-regulator" to avoid misclassification as a canonical transcription factor.

Discussion:

Consolidate repetitive points about TTP expression patterns into a more concise summary.

Replace phrasing like "excessive TTP activates" with more measured terms such as "elevated TTP expression promotes."

Add a sentence clarifying the cell-type-specific logic: e.g., "Although TTP is predominantly hepatocytic, its effect on HSC activation is mediated via paracrine TGF-β1 signaling."

Ensure the tone in the therapeutic implications section is suitably cautious.

PLOS authors have the option to publish the peer review history of their article (what does this mean?). If published, this will include your full peer review and any attached files.

Reviewer #1: No

Reviewer #2: No

Reviewer #3: **Yes:** Xue-nong Luo

**Figure resubmission:**
---

## [Editor Report · Decision Letter 2]

16 Feb 2026

Dear Zhou,

We are pleased to inform you that your manuscript ’Tristetraprolin attenuates schistosomiasis-induced liver fibrosis through m⁶A-mediated regulation of TGF-β1 mRNA stability’ has been provisionally accepted for publication in PLOS Pathogens.

Should you, your institution’s press office or the journal office choose to press release your paper, you will automatically be opted out of early publication. We ask that you notify us now if you or your institution is planning to press release the article. All press must be co-ordinated with PLOS.

Best regards,

David L. Williams

Academic Editor

PLOS Pathogens

Edward Mitre

Section Editor

PLOS Pathogens

Sumita Bhaduri-McIntosh

Editor-in-Chief

PLOS Pathogens

orcid.org/0000-0003-2946-9497

Michael Malim

Editor-in-Chief

PLOS Pathogens

orcid.org/0000-0002-7699-2064
---

## [Editor Report · Acceptance letter]

Dear Dr Zhou,

We are delighted to inform you that your manuscript, "Tristetraprolin attenuates schistosomiasis-induced liver fibrosis through m⁶A-mediated regulation of TGF-β1 mRNA stability," has been formally accepted for publication in PLOS Pathogens.

Soon after your final files are uploaded, the early version of your manuscript, if you opted to have an early version of your article, will be published online. The date of the early version will be your article’s publication date. The final article will be published to the same URL, and all versions of the paper will be accessible to readers.

Best regards,

Sumita Bhaduri-McIntosh

Editor-in-Chief

PLOS Pathogens

orcid.org/0000-0003-2946-9497

Michael Malim

Editor-in-Chief

PLOS Pathogens

orcid.org/0000-0002-7699-2064